# In vivo dynamics of skeletal muscle Dystrophin in zebrafish embryos revealed by improved FRAP analysis

Fernanda Bajanca[1,2]*, Vinicio Gonzalez-Perez[3], Sean J Gillespie[3], Cyriaque Beley[4,5], Luis Garcia[4,5], Eric Theveneau[2], Richard P Sear[3]†, Simon M Hughes[1]*†

[1]Randall Division of Cell and Molecular Biophysics, King's College London, London, United Kingdom; [2]Centre de Biologie du Développement, CNRS and Université Paul Sabatier, Toulouse, France; [3]Department of Physics, University of Surrey, Guildford, United Kingdom; [4]Research unit Inserm, Université Versailles Saint-Quentin, Montigny-le-Bretonneux, France; [5]Laboratoire International Associé–Biologie appliquée aux handicaps neuromusculaires, Centre Scientifique de Monaco, Monaco, Monaco

**Abstract** Dystrophin forms an essential link between sarcolemma and cytoskeleton, perturbation of which causes muscular dystrophy. We analysed Dystrophin binding dynamics in vivo for the first time. Within maturing fibres of host zebrafish embryos, our analysis reveals a pool of diffusible Dystrophin and complexes bound at the fibre membrane. Combining modelling, an improved FRAP methodology and direct semi-quantitative analysis of bleaching suggests the existence of two membrane-bound Dystrophin populations with widely differing bound lifetimes: a stable, tightly bound pool, and a dynamic bound pool with high turnover rate that exchanges with the cytoplasmic pool. The three populations were found consistently in human and zebrafish Dystrophins overexpressed in wild-type or *dmd^{ta222a/ta222a}* zebrafish embryos, which lack Dystrophin, and in *Gt (dmd-Citrine)^{ct90a}* that express endogenously-driven tagged zebrafish Dystrophin. These results lead to a new model for Dystrophin membrane association in developing muscle, and highlight our methodology as a valuable strategy for in vivo analysis of complex protein dynamics.

*For correspondence:
fernanda.vinagre-bajanca@univ-tlse3.fr (FB); simon.hughes@kcl.ac.uk (SMH)

†These authors contributed equally to this work

## Introduction

Muscle Dystrophin establishes a link between Dystroglycan complexes at the cell membrane and actin in the cortical cytoskeleton (*Ibraghimov-Beskrovnaya et al., 1992*; *Levine et al., 1992*; *Ervasti and Campbell, 1993*; *Rybakova et al., 1996*, *2000*). Mutations in the *Dystrophin* gene often lead to a non-functional protein and Duchenne muscular dystrophy (DMD), characterised by severe muscle degeneration from early childhood. In-frame deletions within the Dystrophin sequence can result in a shortened but partially functional protein that causes Becker muscular dystrophy (BMD) (*Koenig et al., 1989*).

A major international effort aims to develop gene therapy for DMD. Yet, there are still big gaps on our understanding of how Dystrophin works within cells. It is important to understand the dynamics of Dystrophin in vivo and how this could vary within cellular context, influencing the phenotype of BMD and gene therapy planning for patients with DMD. For example, many current approaches for gene therapy in DMD aim to restore 'short' Dystrophins, known to be partially functional from studies of patients with BMD and murine transgenic models (*Konieczny et al., 2013*). How the dynamics of these proteins compare with those of full-length Dystrophin has not been addressed due to the lack of a suitable method. However, if some short Dystrophin forms bind more efficiently and stably than

**eLife digest** A protein called Dystrophin plays a key role in maintaining the structural integrity of muscle cells as they contract and relax. Mutations in the gene that encodes Dystrophin can cause several different types of muscular dystrophy, a group of diseases in which muscle progressively weakens. Some mutations in Dystrophin can lead to mild symptoms that may affect the quality of life but are not life threatening. However, in more serious cases, patients lose the ability to walk in childhood and have shortened life expectancy. There is no cure for these diseases, and there are still big gaps in our understanding of how Dystrophin works, which makes it more difficult to develop efficient therapies.

The zebrafish is often used as a model to study muscular dystrophies. In this study, Bajanca et al. introduced human Dystrophin into zebrafish muscle cells and analysed its behaviour using a combination of mathematical modelling and a method known as 'fluorescence recovery after photobleaching'. In these experiments, the human Dystrophin was attached to a tag that fluoresces green under a microscope, which allowed it to be easily seen and be followed in real time inside the cells of live animals.

Bajanca et al. observed that Dystrophin could either remain firmly associated with the membrane that surrounds the cell over long periods of time or interact briefly with the membrane. Bajanca et al. carried out further experiments with the Dystrophin protein naturally found in zebrafish and observed that it behaved in a similar manner to the human protein, suggesting this behaviour is likely to be important for the ability of the protein to work.

Bajanca et al.'s findings reveal that Dystrophin displays complex behaviour in living muscle cells. The fact that some Dystrophin molecules are firmly attached to the membrane support previous findings that this protein provides mechanical stability to the cells. However, the discovery that there is a group of more mobile Dystrophin molecules within muscle cells suggests that this protein may also play other roles. Therefore, these findings open a new avenue for research that may contribute to the development of new therapy approaches in future.

others this will have an impact on the relative amount of protein necessary to recover function. The knowledge of Dystrophin dynamics and a methodology to perform comparative studies is therefore needed.

Dystrophin is well studied in zebrafish and its homology with the human Dystrophin is well documented (*Guyon et al, 2003*; *Jin et al., 2007*; *Berger et al., 2011*; *Lai et al., 2012*). Several mutant and transgenic lines have been used as model for Duchenne muscular dystrophy and testing potential therapeutic targets (*Kunkel et al., 2006*; *Johnson et al., 2013*; *Kawahara and Kunkel, 2013*; *Waugh et al., 2014*; *Wood and Currie, 2014*). The loss of Dystrophin is lethal to both people and zebrafish, primarily due to striated muscle defects (*Bassett et al., 2003*; *Berger et al., 2010*). Both species show developmental progression towards the adult localisation of Dystrophin. In human embryos, Dystrophin first appears in the cytoplasm, at the tips of myotubes, then becomes widespread throughout the myofibres in foetal stages (*Wessels et al., 1991*; *Clerk et al., 1992*; *Chevron et al., 1994*; *Mora et al., 1996*; *Torelli et al., 1999*). In embryonic zebrafish muscle, Dystrophin transcripts are reported to accumulate initially in the cytoplasm, and from 24 hr post fertilization (hpf) until early larval stages, Dystrophin protein and transcripts are primarily located at muscle fibre tips (*Bassett et al., 2003*; *Guyon et al., 2003*; *Jin et al., 2007*; *Böhm et al., 2008*; *Ruf-Zamojski et al., 2015*). In both species, Dystrophin becomes localised under the sarcolemma in maturing and adult muscle fibres where it concentrates at costameres, neuromuscular and myotendinous junctions (*Samitt and Bonilla, 1990*; *Miyatake et al., 1991*; *Chambers et al., 2001*; *Guyon et al., 2003*). Dystrophin half-life is believed to be very long (*Tennyson et al., 1996*; *Verhaart et al., 2014*). Therefore, to study Dystrophin binding dynamics, it may be advantageous to look at the moment where binding complexes are actively forming, during muscle development.

Study of protein dynamics in living tissue faces many technical hurdles that no available method can tackle satisfactorily. Fluorescence correlation spectroscopy (FCS) requires stable confocal imaging of submicron volumes and is thus sensitive to drift in living tissue. Moreover, FCS is only applicable over a limited range of fluorophore concentrations and is greatly impeded by the presence of significant

quantities of immobile fluorophores. Fluorescence recovery after photobleaching (FRAP) avoids these problems. However, imaging in a living organism is challenging due to low signal-to-noise ratio that worsens as tissue thickness increases and protein abundance decreases. In addition, cells are located at variable optical depths and have varying shapes and protein levels, all of which introduces variability. This hampers identification of real variation in protein dynamics and prevents the common procedure of pooling data from multiple cells to reduce noise.

In this study, we assess human Dystrophin dynamics in muscle cells of host zebrafish embryos, using a new approach to perform and analyse FRAP in the context of the living muscle fibre that specifically deals with the challenges of in vivo protein analysis. We thoroughly characterize the expression of the exogenous human Dystrophin within zebrafish host muscle cells. Overexpression often results in an excess of cytoplasmic Dystrophin, which is taken into account on the analysis of Dystrophin binding dynamics. We demonstrate that Dystrophin diffuses freely in the zebrafish muscle fibre cytoplasm and determine its diffusion constant. At the binding sites localised at the muscle cell tips, we found the existence of two membrane-bound pools with distinct binding constants: an immobile pool bound stably during our imaging timescale and a mobile-bound pool with a highly dynamic turnover. We test several potential factors that could potentially interfere with the binding dynamics of Dystrophin, or with its analysis, and result in wrong identification of a labile-bound pool: lateral diffusion of bound Dystrophin, transient dark state of fluorescent proteins, artificial increase of the cytoplasmic pool, competition with endogenous zebrafish Dystrophin, or weak interaction between inter-species proteins. Our data allowed us to dismiss all these hypotheses, supporting the real existence of two bound forms of Dystrophin in maturing fibres of the zebrafish embryo. Taken together, these results suggest a model for Dystrophin association with the membrane and provide a baseline and a validated methodology to analyse how modifications in Dystrophin structure may alter its dynamics.

## Results

### Dystrophin mRNA and protein localization are environmentally determined

We set out to analyse human Dystrophin protein dynamics in vivo in the physiological environment of the muscle fibres of zebrafish embryos (*Figure 1*). We engineered expression constructs based on the full-length 427 kd human cDNA sequence (huDys; *Figure 1A*; 'Materials and methods'). Expression of huDys or GFP control in zebrafish embryos was achieved through the injection of the DNA constructs into newly fertilized embryos at the early 1 cell stage, aiming to obtain mosaic expression to facilitate single cell analysis (*Figure 1B,C*). From 24 hpf onwards, huDys (*Figure 1C*, green) accumulated progressively at both ends of transgenic fibres (hereafter referred to as 'tips'), as observed for endogenous zebrafish Dystrophin (*Figure 1C*, red). GFP control showed no tip accumulation (*Figure 1D*). In addition, huDys was often detected accumulating at putative neuromuscular junctions (NMJ), like endogenous Dystrophin (arrows in *Figure 1E,F*). We conclude that human Dystrophin localises in zebrafish skeletal muscle like zebrafish Dystrophin, making it likely, in a first approach, that the zebrafish embryo could be a suitable host to study human Dystrophin in vivo.

To allow the in vivo study of huDys dynamics, the expression construct was modified to produce huDys tagged with GFP at its C-terminus (huDysGFP; *Figure 1A*; 'Materials and methods'). This produces a bright fluorescent signal easily detectable at fibre tips (*Figure 1G*, arrows). Occasionally, some cells showed accumulations at membrane protrusions (*Figure 1G*, yellow arrowheads) and NMJs (*Figure 1G*, red arrowhead). The latter was confirmed by double staining with α-bungarotoxin (*Figure 1H*, inset). Compared to GFP alone, huDysGFP was generally less bright (*Figure 2A*) but was, nevertheless, more readily detected in muscle than non-muscle tissue (*Figure 2B*), suggesting that binding and stabilization at the membrane differ between tissues.

To determine whether human *Dystrophin* mRNA becomes localised in zebrafish muscle like the endogenous transcripts, in situ mRNA hybridization with a human *Dystrophin*-specific probe was performed on injected embryos. In most cases, localisation of human *Dystrophin* mRNA was observed at fibre tips (*Figure 1I,J*). Thus, GFP-tagged Dystrophin localises similarly to its untagged counterpart, and to the endogenous Dystrophin mRNA and protein, and it is suitable for in vivo imaging.

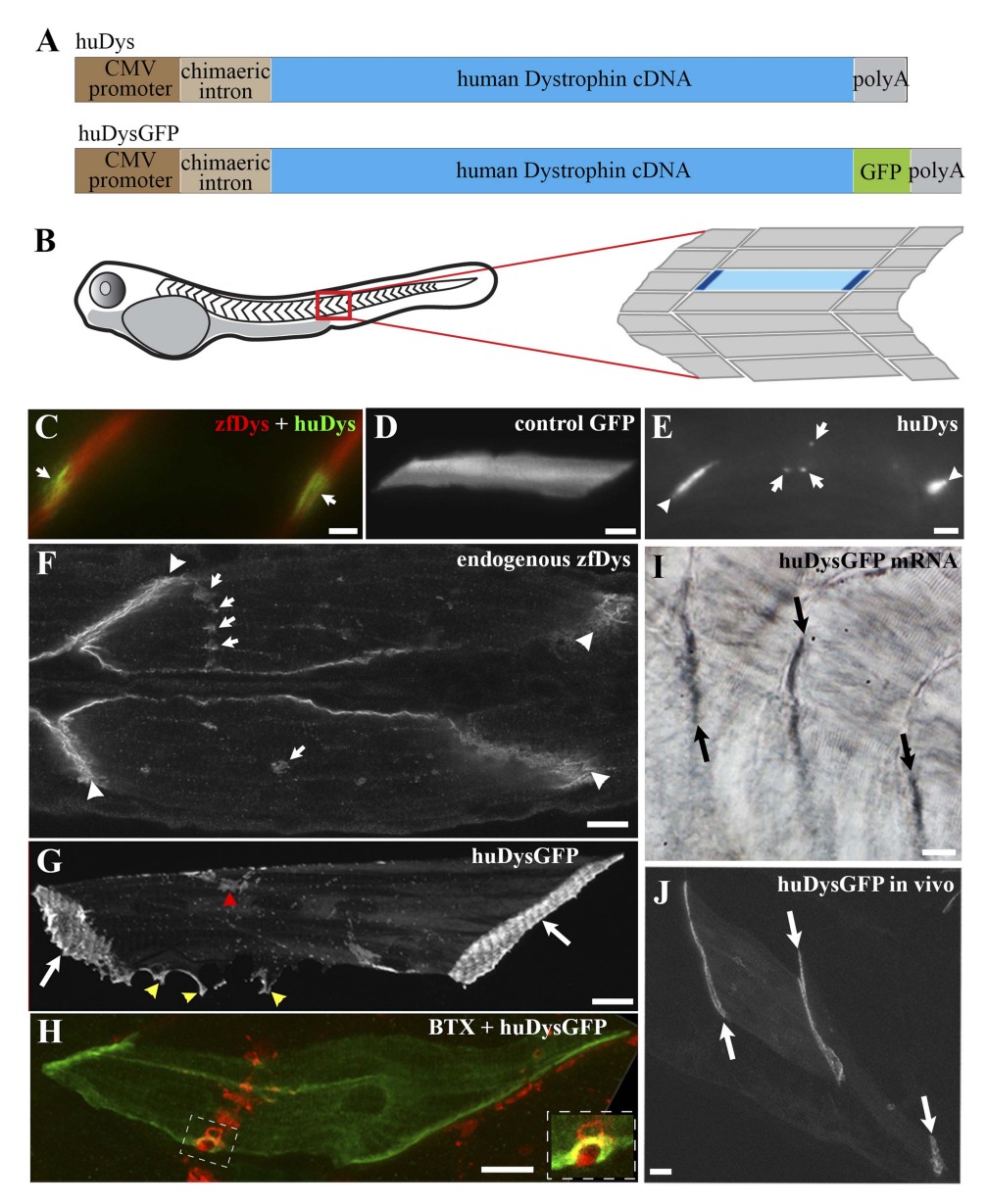

**Figure 1**. Human Dystrophin expression in the zebrafish embryo. (**A**) Main features of the human Dystrophin expression constructs engineered for this study. (**B**) Schematic illustrating 2 dpf zebrafish embryo. Slow muscle fibres within the chevron-shaped somite, one magnified and highlighted in blue, are typically aligned anterior-posteriorly with their tips (dark blue) attaching at vertical somite borders. (**C**) Immunofluorescent detection of exogenous huDys (green, arrows) at fibre tips, co-localizing with endogenous zebrafish Dystrophin (zfDys, red) that accumulates at the tips of every muscle fibre, marking the somite border. (**D**) In vivo expression of control GFP shows accumulation in muscle fibre cytoplasm without enrichment at the fibre tips. (**E**) Immunodetection with antibody specifically recognizing human Dystrophin on whole mount 2 dpf embryo shows punctate accumulation of exogenous huDys (arrow) suggestive of localization at the NMJ, in addition to fibre tips (arrowheads). (**F**) Immunodetection on longitudinal cryostat sections of 2 dpf somitic muscle shows enrichment of endogenous zebrafish Dystrophin (zfDys) at NMJ (arrows). Note concentration of most zfDys at fibre tips (arrowheads). (**G**) Maximum intensity projection of a confocal stack showing accumulation of huDysGFP in a muscle fibre in vivo. Strong enrichment is noticeable at the tips (arrows), membrane protrusions (yellow arrowheads), and NMJ (red arrowheads). (**H**) Double immunofluorescent detection of GFP in a huDysGFP-expressing embryo (huDysGFP, green) and α-bungarotoxin (BTX, red) confirms co-localization at the NMJ (insert). (**I**, **J**) huDysGFP mRNA detected by in situ hybridization (arrows in **I**, Nomarski) localises at fibre tips like GFP fluorescence detected while in vivo (arrows in **J**; confocal maximum projection). Scale bars = 10 µm.

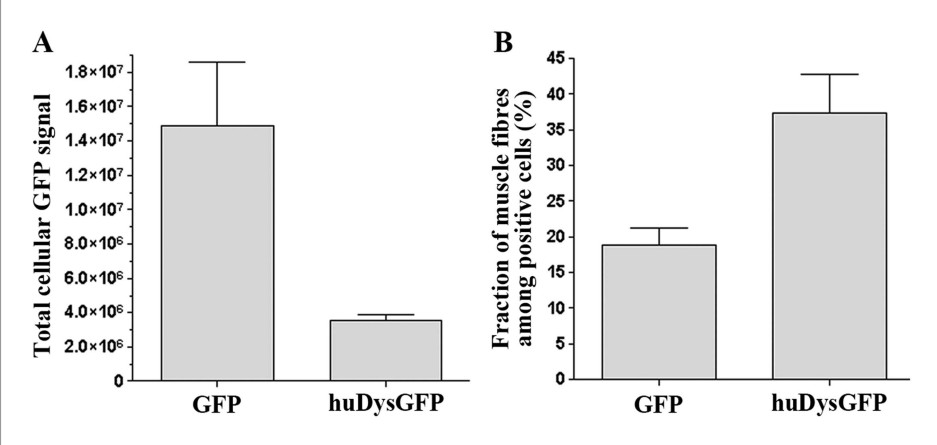

**Figure 2**. Comparison of huDysGFP and GFP expression in 2 dpf zebrafish embryos. (**A**) Total cellular GFP signal (sum of pixel values) of sum projections made from confocal optical sections of individual muscle fibres expressing GFP or huDysGFP in vivo. $N_{GFP}$ = 10 fibres, $N_{huDysGFP}$ = 32 fibres; p < 0. 0001. (**B**) Fraction of muscle fibres among positive cells in embryos expressing huDysGFP or GFP in vivo. $N_{GFP}$ = 1593 cells in 27 embryos, $N_{huDysGFP}$ = 472 cells in 28 embryos; p = 0. 0032. Error bars show S.E.M.

## Increase of cytoplasmic Dystrophin does not affect accumulation at the fibre tips

Both endogenous Dystrophin and huDysGFP accumulate at the fibre tips, yet the endogenous form is not readily detected in cytoplasm in immunofluorescence assays, that is, it is not clear that the fluorescence detected is higher than background (*Figure 1F*), in contrast most fibres expressing huDysGFP show weak but detectable fluorescence in the cytoplasm (*Figure 1G,H*). We investigated this difference.

As intensity around 3 units above the background is easily detected under our imaging conditions ('Materials and methods'), we can distinguish huDysGFP in a cytoplasmic voxel (a three dimensional pixel of 0.024 $\mu m^3$) down to a number per voxel around 60 times lower than in the brightest fibre tip voxel (avoiding saturation of the detector by setting it to under 255 on 8-bit grayscale). As less than 1% of the entire cell volume is in the tip region, it is possible that even in cells with cytoplasmic huDysGFP below detectable levels there could be as much huDysGFP in the cytoplasm as in the tip region. This could equally be the case for endogenous Dystrophin. Therefore, the observed difference may be partially due to lower sensitivity of the antibody detection of cytoplasmic endogenous Dystrophin compared to the higher sensitivity of GFP detection. However, higher levels of cytoplasmic accumulation are likely an artefact of the overexpression of exogenous Dystrophin. Therefore, to confidently analyse Dystrophin binding dynamics, the presence of this cytoplasmic pool has to be taken into account and a deeper characterisation is required.

We analysed in more detail how each pool, tips, and cytoplasm, distribute. As predicted, in the majority of the muscle fibres, most huDysGFP is in the cytoplasm, with only a minority at the tips (*Figure 3A*), even though the higher concentration at the tips might have suggested otherwise (*Figure 1G,H*). Even in cells with cytoplasmic levels close to the detection limit, there is at least as much huDysGFP dispersed in the whole cytoplasm as that concentrated at the tips (*Figure 3A*). Across a population of fibres, more huDysGFP fluorescence was detected in the cytoplasm of fibres with higher total huDysGFP levels (blue triangles in *Figure 3A*). In contrast, the fluorescence at the tips does not increase with the total fluorescence of the fibre (green circles in *Figure 3A*), indicating that tip binding is limited by the presence of a limited number of binding sites that easily saturate. Thus, the accumulation of huDysGFP in the fibre cytoplasm does not appear to affect the binding at the tips. Also, fibre tips generally had greater fluorescence intensity than fibre cytoplasm (*Figure 3B,C*). High intensities at the tips can be achieved even with low cytoplasmic huDysGFP concentrations (*Figure 3B*). Moreover, at low overall fibre intensities, there is clear preference for accumulation at the tips (*Figure 3C*). All these data indicate that human Dystrophin is preferentially bound at the zebrafish fibre tips regardless of the amount of cytoplasmic Dystrophin.

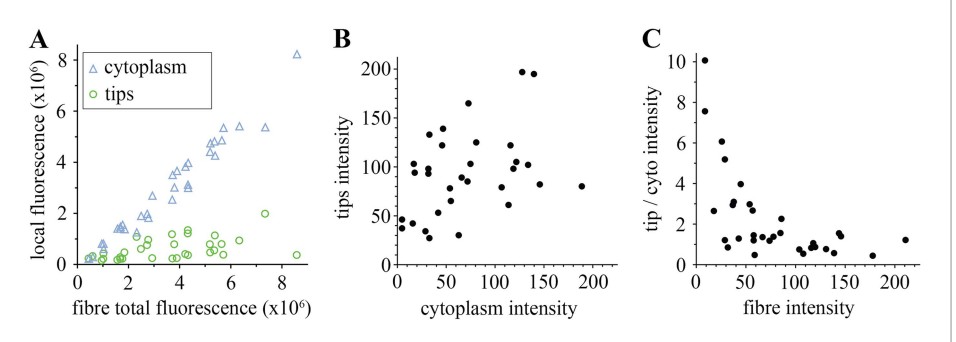

**Figure 3**. Comparison of tip and cytoplasm huDysGFP. (**A**) Variation of the total fluorescence (as raw integrated density or sum of pixel values) at tips (green circles) and cytoplasm (blue triangles) over a population of 32 fibres expressing huDysGFP. (**B**) Mean voxel intensity at tips versus cytoplasm in sum projection of confocal z-stacks. The mean voxel intensity is calculated as the integrated density per pixel in the sum projections, therefore taking into account the fibre and tip size. (**C**) Ratio tip/cytoplasm voxel intensities shows an inverse correlation with the total fibre voxel intensity in sum projections of confocal z-stacks. p value = 0.0004, $R^2$ = 0.3443, n = 32.

## Dystrophin diffuses freely in the cytoplasm

To study Dystrophin dynamics in our system, we still have to take into account the presence of a cytoplasmic pool. Dystrophin is a high molecular mass protein with multiple actin binding sites. We asked whether huDysGFP can diffuse freely in muscle fibre cytoplasm or whether it may be bound to cytoplasmic structures such as actin fibres.

We developed a modified FRAP approach to analyse protein dynamics in vivo. Even in the best imaging conditions, one faces the challenge of low signal-to-noise ratio that worsens as tissue thickness increases and protein abundance decreases. Although we are able to detect cytoplasmic huDysGFP, the signal is weak and the signal-to-noise ratio in single pixels or even small volumes is low (*Figure 4A*). To address these issues, we increase the signal by increasing laser power and studying large areas ('Materials and methods'). Under these conditions, we acquire a consistent signal both for GFP and huDysGFP. However, using a high laser power to image over large areas results in significant photobleaching during imaging, and, even at 100% laser power, bleaching is so slow that significant diffusion occurs during bleaching (*Weiss, 2004*). We compensate for both using a mathematical model applied to the FRAP experimental data ('Materials and methods'). This was integrated in a user-friendly application written to allow easy data analysis of multiple experiments (see 'Materials and methods').

To validate our experimental conditions and FRAP analysis method in the muscle cells of zebrafish embryos, we first analysed GFP diffusion within the cytoplasm. We studied diffusion along the long axis of the muscle cell (*Figure 4B*). The selected fibre is oriented such that the image *X*-axis aligns to the fibre long axis (i.e., roughly anterior-posterior in the animal) and the *Y*-axis is dorso-ventral (*Figure 4C–F*). In each cell, one or two large rectangles of different sizes (narrow and wide) were bleached, avoiding nuclei (*Figure 4B*). The bleached rectangles are shorter along the *X*-axis than in *Y* and cross the entire cell transversely. This makes recovery almost entirely due to mobility along the *X*-axis, simplifying modelling and fitting. We analyse the profile along the *X*-axis taken immediately after bleaching GFP-expressing fibres (*Figure 5A*). This profile is not a top-hat but a Gaussian, much wider than the region actually bleached, consistent with diffusion during the bleaching phase. This diffusion is taken into account in the modelling as neglecting it would lead to significant error (*Castle et al., 2011*; *González-Pérez et al., 2011*; *Müller et al., 2012*) ('Materials and methods').

We fit the recovery for each individual case to obtain the diffusion constant (*D*) and a bleaching-due-to-imaging parameter (*β*) (*Figure 5B*). We tested modelling of the recovery at short and long times post-bleaching. When analysing long (≥10 s) timescales, it is essential to take *β* into account, while for short times bleaching due to imaging is insignificant (*Figure 5B*, *Table 1*). The Gaussian profile and the ability to fit a simple diffusion model to the recovery are strong evidence for GFP diffusion. $D_{GFP}$ best-fit values (fitted to data over ~40 s) averaged 13.2 μm² s⁻¹, ranging from 8.6 to

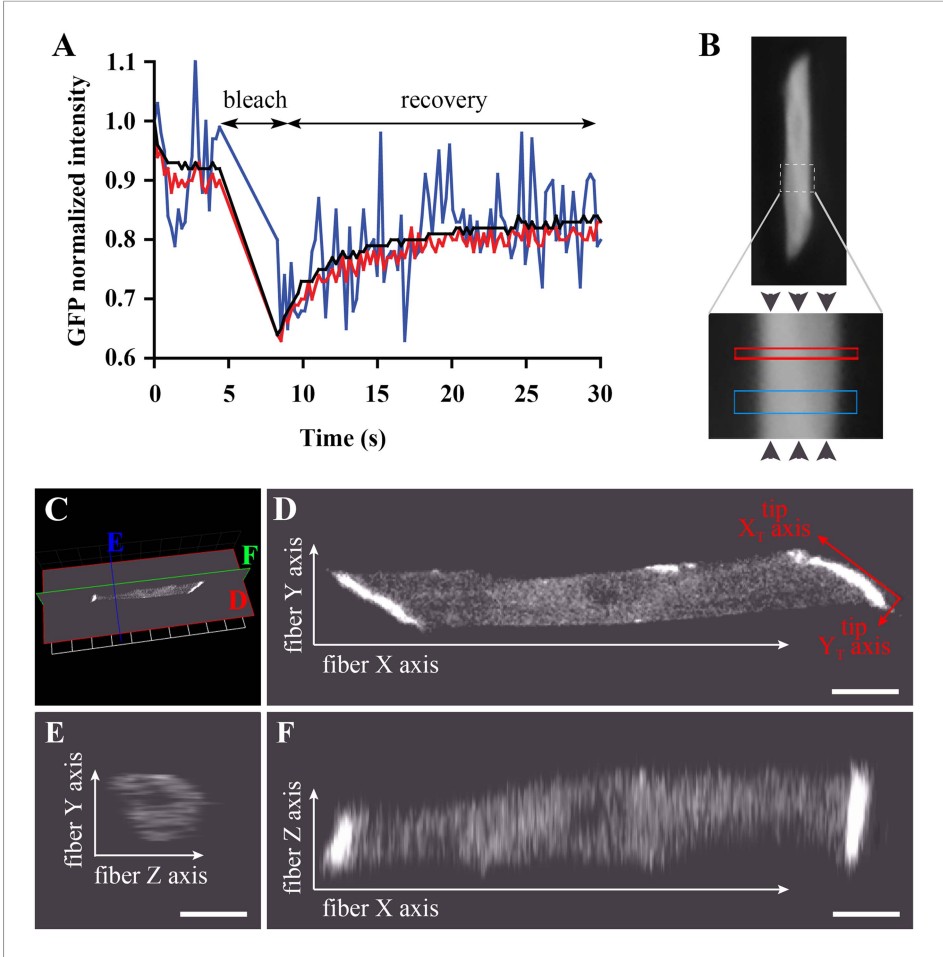

**Figure 4**. Bleaching areas: size optimization, orientation, and definition of Cartesian coordinates. (**A**) Noise reduction in GFP FRAP curve for increasing areas: 1 (blue), 25 (red) and 256 (black) pixels. (**B**) To determine $D$ along the $X$-axis (arrowheads) of individual muscle cells in the embryo, two large regions of different widths (narrow and wide; see *Table 1*) are bleached sequentially and separated by >1 min to ensure full recovery. (**C**) $XYZ$ view from Volocity of a typical muscle fibre expressing huDysGFP in vivo. The cut planes shown correspond to panels **D–F**. (**D**) Muscle fibre imaged in the $XY$ plane from a lateral position of the zebrafish embryo as embedded for FRAP. When referring to the fibre tips, we use a different set of axes: the long axis of the tip is the $X_T$ axis and the shorter is the $Y_T$ axis. (**E**, **F**) $YZ$ and $XZ$ sections, respectively, through the muscle fibre, showing lower $Z$ resolution. Scale bars = 10 µm.

20.8 µm² s⁻¹ (*Table 1*). This matches the range 7.6–15.8 µm² s⁻¹ previously reported for muscle cells (*Arrio-Dupont et al., 2000*; *Kinsey et al., 2011*), which indicates that our approach for in vivo FRAP analysis is able to achieve similar results to those previously reported for isolated cells in culture.

We used the FRAP method described above to analyse cytoplasmic huDysGFP dynamics. The Dystrophin profile immediately after bleaching, similar to that of GFP, is not a top-hat but a Gaussian, wider than the region actually bleached, and consistent with diffusion during the bleaching phase (*Figure 5C*). However, the Gaussian's amplitude is narrower and the depth greater for huDysGFP than GFP, consistent with slower diffusion (compare *Figure 5A,C*). Indeed, $D_{huDysGFP}$ best-fit values (fitted to data over ~40 s) ranged from 1.4 to 10.1 µm² s⁻¹, with a mean of 4.4 µm² s⁻¹ (*Figure 5D*; *Table 1*). Again, the Gaussian profile and our ability to fit a simple diffusion model to the recovery are strong evidence for huDysGFP diffusion, and rule out large scale (above µm) directed motion along the long axis of the cell, or significant binding of huDysGFP to immobile structures.

huDysGFP has significantly lower $D$ than GFP, reflecting the different protein sizes of 454 kD and 27 kD, respectively (*Figure 5E*, p < 0.0001). By comparing narrow and wide bleaches in the same cell across our population of muscle fibres, we found that $D_{huDysGFP}$ appears to vary between cells, that is,

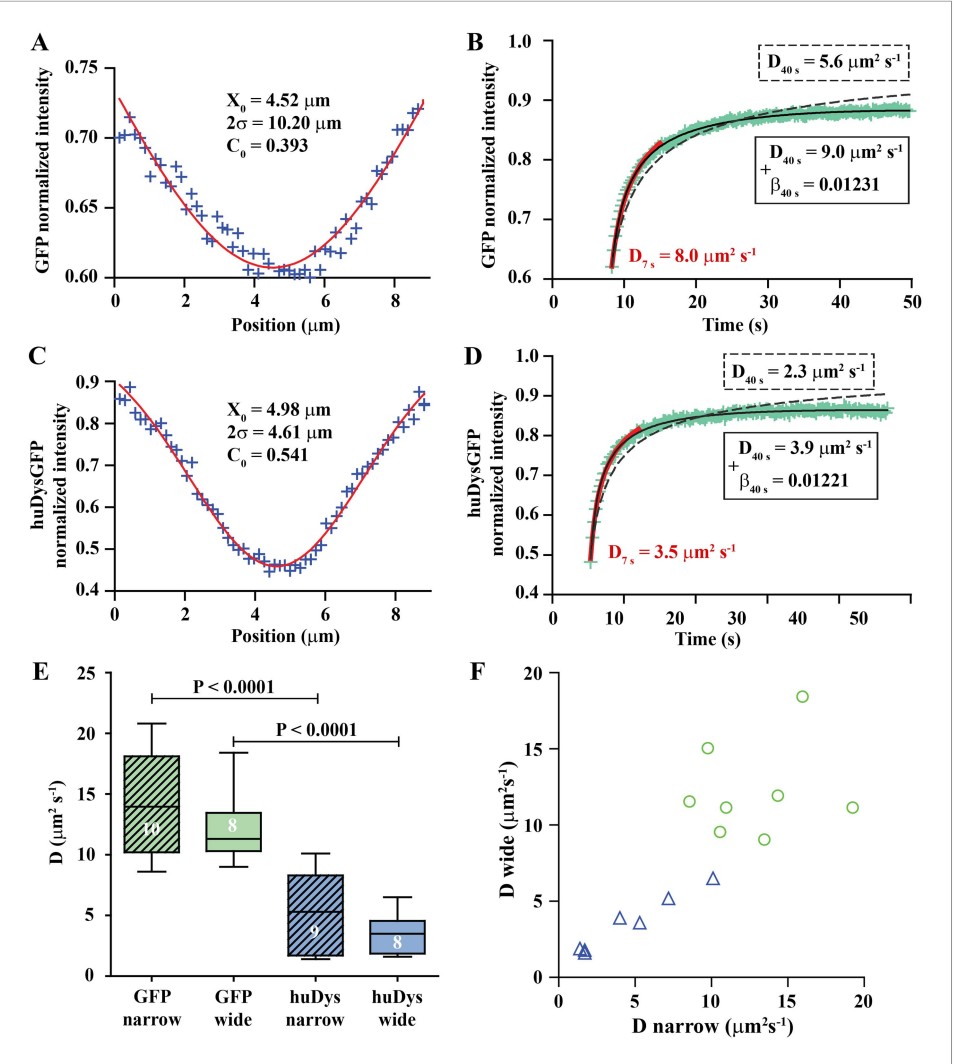

**Figure 5**. Analysis of cytoplasmic diffusion. (**A**–**D**) FRAP experimental data and fitting curves. Normalized intensity profile along the *X*-axis of GFP (**A**) or huDysGFP (**C**) at the first time point after bleaching (blue crosses) and Gaussian fits (red curves). Recovery curves for GFP (**B**) and huDysGFP (**D**) along *X*-axis in which the cyan crosses show normalized fluorescence intensity in the bleached region. Curves are fits of the diffusion model to ∼7 s (red) or ∼40 s (two-parameters, solid black; one-parameter, dashed) post-bleach. (**E**) $D_{GFP}$ and $D_{huDysGFP}$ obtained from two-parameter fits to ∼40 s of FRAP experimental data (see *Table 1*). Graph shows median, quartile, range, and *n*. Comparison was by two tailed *t*-test after test for normality. (**F**) Scatter plot of *D* values obtained for pairs of bleaching experiments performed in the same cell. For huDysGFP (triangles), the two *D* values measured in the same cell, in areas of different widths (narrow/wide), show a good correlation (triangles; Pearson R = 0.98). The triangles are not far from falling on a straight line of slope one. The small cell-to-cell variation in huDysGFP, relative to the variation between cells suggests that the mobility of huDysGFP genuinely varies from one fibre to another. For GFP, there is no definite trend visible, just scatter, presumably due to lower signal-to-noise with the more rapid diffusion of GFP. It is, therefore, not clear whether the mobility of GFP varies significantly from one fibre to another.

there is real variability in huDysGFP cytoplasmic dynamics from one cell to another. The difference between $D_{huDysGFP}$ values for a pair of experiments in the same cell is significantly smaller than between values in different cells (*Figure 5E,F*; *Table 1*). Pairs of *D* values obtained from the same cell are well correlated ($R_{huDysGFP}$ = 0.9813; *Figure 5F*). This indicates that Dystrophin's diffusion in the cytoplasm shows important variations from cell to cell but that it is consistent within one cell. Further, this finding highlights the limitations of reducing noise by 'pooling' FRAP results from more than one cell. Therefore, each FRAP curve was analysed independently throughout this study and pooling was

deliberately avoided. The original data and relevant analysis files for each case are available online (*Bajanca et al., 2015*) to complement representative examples shown in figures and main data summarised in tables. We conclude that, although there is fibre-specific variation of $D$, there is no evidence that cytoplasmic Dystrophin either binds cytoskeletal elements or is actively transported towards fibre tips.

## Human Dystrophin is bound at muscle fibre tips

The fibre tip region presumably contains huDysGFP bound to the Dystroglycan complex at the membrane (*Guyon et al., 2003*; *Böhm et al., 2008*) and a portion diffusing in the adjacent cytoplasm. FRAP can test whether huDysGFP is able to bind stably at the tips and reveal binding dynamics, but this is a challenging task. Diffusion of cytoplasmic protein into the bleached area, combined with bleaching-due-to-imaging, masks the real binding dynamics. We tested diffusion-plus-binding models to analyse the tip FRAP curves but these require too many variables to be reliably fitted, reflecting the complexity of the binding dynamics, as will be shown. The relatively featureless, and noisy, recovery curves of in vivo FRAP within a complex tissue cannot adequately constrain this number of variables (*Sadegh Zadeh et al., 2006*). Instead, to understand human Dystrophin dynamics at the tip, we use direct semi-quantitative analysis of bleaching and recovery. A FRAP protocol that bleached only part of the tip allowed comparison of bleached and unbleached regions of the tip (*Figure 6A*). We combined this with analysis of the cytoplasm intensity near the tip in each cell to measure how much recovery to expect due to diffusion. In addition, recovery is followed at an initial fast acquisition rate to detect fast recovery, and then at a slower pace over longer time periods with reduced bleaching-due-to-imaging (*Figure 6A*).

On most tips, we identified three signatures of a population of Dystrophin that is bound at the tip and immobile (not turning over) on the timescale of our experiment. First, there is only partial recovery in the bleached region of the tip regardless of the huDysGFP expression levels (*Figure 6B–F*). Second, the normalized intensity difference between the unbleached and bleached regions in most tips (black crosses in *Figure 6C–E*) reaches a plateau above zero (*Table 2*). At the end of the experiment, both halves of the tip have received the same amount of light (and thus bleaching) for ~250 s after the intentional bleaching phase, so the difference observed is created by the huDysGFP bleached earlier that did not turn over. Third, in the unbleached half of the tip, which receives the same bleaching-due-to-imaging as the cytoplasm, both the normalized and the absolute drops in intensity from start to end of the experiment are much larger than in the cytoplasm (compare drops in the tips with those in the cytoplasms in *Figure 6F*). Thus, our FRAP data consistently confirm that a population of huDysGFP is effectively bound and immobile at zebrafish muscle fibre tips.

## Two bound populations of human Dystrophin at muscle fibre tips

We analysed the dynamics of exchange of cytoplasmic freely diffusing huDysGFP with bound huDysGFP. Surprisingly, the FRAP curves revealed complex binding dynamics of Dystrophin at the tips that vary between fibres. We analysed the recovery pattern in the bleached region when switching to a slow acquisition regimen (transition from step 3 to 4 in *Figure 6A*). Despite the very slow turnover that characterizes the immobile-bound pool at the tips, for many fibres there is an almost immediate partial recovery (see *Figure 6C–E*). A shift of the dynamic equilibrium between dark-state and excitable GFP could result in apparent recovery following a switch to slow acquisition speed (*Mueller et al., 2012*). However, we verified that the extent to which this phenomenon occurs in huDysGFP could only justify a very small fraction of recovery (<1%, *Figure 7A*). We next evaluated the contribution of free diffusion of cytoplasmic huDysGFP into the bleached region, occurring as demonstrated with a $D$ around 4 $\mu m^2$ $s^{-1}$. As the bleached region contains bound and unbound huDysGFP and both are subjected to bleaching, it is not straightforward to analyse the recovery against a control cytoplasmic region, which is not subjected to intentional bleaching. Therefore, we compared unbleached tip and cytoplasm traces, which were submitted to comparable experimental conditions, without intentional bleaching but with strong bleaching-due-to-imaging during the fast acquisition phase (*Figure 6F*). On switching to slow acquisition, any recovery on the unbleached tip would be expected to parallel the recovery on the cytoplasm. Nevertheless, the immediate recovery is in most cases larger than that expected from diffusion of cytoplasmic huDysGFP despite a steady state is reached soon after, characteristic of the large immobile pool (*Figure 6F*, compare open or

**Table 1**. Diffusion constants, *D*, for GFP and Dystrophin in the cytoplasm obtained from fitting to FRAP experimental data

| Data set number (cell number) | Bleach width (pixels) | Final fitted time point | Fibre length (µm) | Bleach position (µm) | *D* (µm/s²) | β |
|---|---|---|---|---|---|---|
| GFP, standard fibre length | | | | | | |
| set 1 (cell 1) | 10 | 500 | 90.0 | 45.0 | 9.3 | 0.001753 |
| set 2 (cell 1) | 20 | 500 | 90.0 | 45.0 | 8.2 | 0.001572 |
| set 3 (cell 2) | 10 | 500 | 90.0 | 45.0 | 9.6 | 0.001502 |
| set 4 (cell 2) | 20 | 500 | 90.0 | 45.0 | 9.6 | 0.001483 |
| set 5 (cell 3) | 10 | 500 | 90.0 | 45.0 | 6.3 | 0.002618 |
| set 6 (cell 3) | 20 | 500 | 90.0 | 45.0 | 7.3 | 0.002503 |
| set 7 (cell 4) | 10 | 500 | 90.0 | 45.0 | 10.2 | 0.001980 |
| set 8 (cell 4) | 20 | 500 | 90.0 | 45.0 | 12.8 | 0.001181 |
| set 9 (cell 5) | 10 | 500 | 90.0 | 45.0 | 12.4 | 0.002358 |
| set 10 (cell 5) | 20 | 500 | 90.0 | 45.0 | 10.2 | 0.000961 |
| set 1 (cell 1) | 10 | 200 | 90.0 | 45.0 | 11.0 | 0.002259 |
| set 2 (cell 1) | 20 | 200 | 90.0 | 45.0 | 11.1 | 0.002861 |
| set 3 (cell 2) | 10 | 200 | 90.0 | 45.0 | 10.6 | 0.001721 |
| set 4 (cell 2) | 20 | 200 | 90.0 | 45.0 | 9.5 | 0.001418 |
| set 5 (cell 3) | 10 | 200 | 90.0 | 45.0 | 8.6 | 0.003959 |
| set 6 (cell 3) | 20 | 200 | 90.0 | 45.0 | 11.5 | 0.004743 |
| set 7 (cell 4) | 10 | 200 | 90.0 | 45.0 | 9.8 | 0.001765 |
| set 8 (cell 4) | 20 | 200 | 90.0 | 45.0 | 15.0 | 0.002000 |
| set 9 (cell 5) | 10 | 200 | 90.0 | 45.0 | 14.4 | 0.002998 |
| set 10 (cell 5) | 20 | 200 | 90.0 | 45.0 | 11.9 | 0.001895 |
| set 11 (cell 6) | 10 | 200 | 90.0 | 45.0 | 16.0 | 0.001196 |
| set 12 (cell 6) | 20 | 200 | 90.0 | 45.0 | 18.4 | 0.002364 |
| set 1 (cell 1) | 10 | 50 | 90.0 | 45.0 | 10.1 | 0.000000 |
| set 2 (cell 1) | 20 | 50 | 90.0 | 45.0 | 9.8 | 0.000000 |
| set 3 (cell 2) | 10 | 50 | 90.0 | 45.0 | 9.4 | 0.000000 |
| set 4 (cell 2) | 20 | 50 | 90.0 | 45.0 | 8.3 | 0.000000 |
| set 5 (cell 3) | 10 | 50 | 90.0 | 45.0 | 7.6 | 0.000000 |
| set 6 (cell 3) | 20 | 50 | 90.0 | 45.0 | 9.9 | 0.000000 |
| set 7 (cell 4) | 10 | 50 | 90.0 | 45.0 | 9.0 | 0.000000 |
| set 8 (cell 4) | 20 | 50 | 90.0 | 45.0 | 13.8 | 0.000000 |
| set 9 (cell 5) | 10 | 50 | 90.0 | 45.0 | 11.5 | 0.000000 |
| set 10 (cell 5) | 20 | 50 | 90.0 | 45.0 | 11.7 | 0.000000 |
| set 11 (cell 6) | 10 | 50 | 90.0 | 45.0 | 15.2 | 0.000000 |
| set 12 (cell 6) | 20 | 50 | 90.0 | 45.0 | 14.5 | 0.000000 |
| GFP, comparing measured and standard fibre length | | | | | | |
| set 13 (cell 7) | 8 | 200 | 82.0 | 41.0 | 13.3 | 0.001050 |
| set 14 (cell 7) | 32 | 200 | 82.0 | 41.0 | 8.9 | 0.001208 |
| set 15 (cell 8) | 8 | 200 | 79.0 | 40.0 | 18.5 | 0.001700 |
| set 16 (cell 8) | 32 | 200 | 79.0 | 40.0 | 10.8 | 0.002270 |
| set 17 (cell 9) | 8 | 200 | 83.0 | 30.0 | 19.8 | 0.000616 |
| set 18 (cell 10) | 8 | 200 | 102.0 | 44.0 | 17.1 | 0.001177 |
| set 13 (cell 7) | 8 | 200 | 90.0 | 45.0 | 13.5 | 0.001084 |

*Table 1. Continued on next page*

Bajanca *et al*. eLife 2015;4:e06541. DOI: 10.7554/eLife.06541

*Table 1. Continued*

| Data set number (cell number) | Bleach width (pixels) | Final fitted time point | Fibre length (µm) | Bleach position (µm) | D (µm/s²) | β |
|---|---|---|---|---|---|---|
| set 14 (cell 7) | 32 | 200 | 90.0 | 45.0 | 9.0 | 0.001231 |
| set 15 (cell 8) | 8 | 200 | 90.0 | 45.0 | 19.3 | 0.001828 |
| set 16 (cell 8) | 32 | 200 | 90.0 | 45.0 | 11.1 | 0.002377 |
| set 17 (cell 9) | 8 | 200 | 90.0 | 45.0 | 20.8 | 0.000828 |
| set 18 (cell 10) | 8 | 200 | 90.0 | 45.0 | 16.9 | 0.001144 |
| set 13 (cell 7) | 8 | 50 | 90.0 | 45.0 | 10.9 | 0.000000 |
| set 14 (cell 7) | 32 | 50 | 90.0 | 45.0 | 8.0 | 0.000000 |
| set 15 (cell 8) | 8 | 50 | 90.0 | 45.0 | 12.9 | 0.000000 |
| set 16 (cell 8) | 32 | 50 | 90.0 | 45.0 | 9.0 | 0.000000 |
| set 17 (cell 9) | 8 | 50 | 90.0 | 45.0 | 18.5 | 0.000000 |
| set 18 (cell 10) | 8 | 50 | 90.0 | 45.0 | 15.2 | 0.000000 |
| huDysGFP, standard fibre length | | | | | | |
| set 19 (cell 11) | 8 | 200 | 90.0 | 45.0 | 10.1 | 0.005649 |
| set 20 (cell 11) | 16 | 200 | 90.0 | 45.0 | 6.5 | 0.005012 |
| set 21 (cell 12) | 8 | 200 | 90.0 | 45.0 | 7.2 | 0.002094 |
| set 22 (cell 12) | 16 | 200 | 90.0 | 45.0 | 5.2 | 0.001708 |
| set 23 (cell 13) | 8 | 200 | 90.0 | 45.0 | 1.7 | 0.003047 |
| set 24 (cell 13) | 16 | 200 | 90.0 | 45.0 | 1.8 | 0.001818 |
| set 25 (cell 14) | 8 | 200 | 90.0 | 45.0 | 9.4 | 0.008203 |
| set 26 (cell 15) | 8 | 200 | 90.0 | 45.0 | 6.0 | 0.004196 |
| set 27 (cell 16) | 4 | 200 | 90.0 | 45.0 | 1.4 | 0.001003 |
| set 28 (cell 16) | 10 | 200 | 90.0 | 45.0 | 1.9 | 0.000966 |
| set 19 (cell 11) | 8 | 50 | 90.0 | 45.0 | 8.7 | 0.000000 |
| set 20 (cell 11) | 16 | 50 | 90.0 | 45.0 | 5.2 | 0.000000 |
| set 21 (cell 12) | 8 | 50 | 90.0 | 45.0 | 6.4 | 0.000000 |
| set 22 (cell 12) | 16 | 50 | 90.0 | 45.0 | 4.9 | 0.000000 |
| set 23 (cell 13) | 8 | 50 | 90.0 | 45.0 | 1.5 | 0.000000 |
| set 24 (cell 13) | 16 | 50 | 90.0 | 45.0 | 1.7 | 0.000000 |
| set 25 (cell 14) | 8 | 50 | 90.0 | 45.0 | 7.7 | 0.000000 |
| set 26 (cell 15) | 8 | 50 | 90.0 | 45.0 | 5.1 | 0.000000 |
| set 27 (cell 16) | 4 | 50 | 90.0 | 45.0 | 1.3 | 0.000000 |
| set 28 (cell 16) | 10 | 50 | 90.0 | 45.0 | 1.8 | 0.000000 |
| huDysGFP, comparing measured and standard fibre length | | | | | | |
| set 29 (cell 17) | 4 | 200 | 130.0 | 62.0 | 1.7 | 0.001319 |
| set 30 (cell 17) | 10 | 200 | 130.0 | 62.0 | 1.6 | 0.001140 |
| set 31 (cell 18) | 4 | 200 | 124.0 | 49.0 | 4.0 | 0.000757 |
| set 32 (cell 18) | 10 | 200 | 124.0 | 49.0 | 3.9 | 0.001221 |
| set 33 (cell 19) | 4 | 200 | 106.0 | 51.0 | 5.3 | 0.001332 |
| set 34 (cell 19) | 10 | 200 | 106.0 | 51.0 | 3.6 | 0.001408 |
| set 35 (cell 20) | 10 | 200 | 112.0 | 31.0 | 3.4 | 0.004444 |
| set 29 (cell 17) | 4 | 200 | 90.0 | 45.0 | 1.7 | 0.001319 |
| set 30 (cell 17) | 10 | 200 | 90.0 | 45.0 | 1.6 | 0.001140 |
| set 31 (cell 18) | 4 | 200 | 90.0 | 45.0 | 4.0 | 0.000757 |

Table 1. Continued

| Data set number (cell number) | Bleach width (pixels) | Final fitted time point | Fibre length (µm) | Bleach position (µm) | D (µm/s²) | β |
|---|---|---|---|---|---|---|
| set 32 (cell 18) | 10 | 200 | 90.0 | 45.0 | 3.9 | 0.001221 |
| set 33 (cell 19) | 4 | 200 | 90.0 | 45.0 | 5.3 | 0.001332 |
| set 34 (cell 19) | 10 | 200 | 90.0 | 45.0 | 3.6 | 0.001408 |
| set 35 (cell 20) | 10 | 200 | 90.0 | 45.0 | 3.4 | 0.004447 |
| set 29 (cell 17) | 4 | 50 | 90.0 | 45.0 | 1.4 | 0.000000 |
| set 30 (cell 17) | 10 | 50 | 90.0 | 45.0 | 1.6 | 0.000000 |
| set 31 (cell 18) | 4 | 50 | 90.0 | 45.0 | 3.6 | 0.000000 |
| set 32 (cell 18) | 10 | 50 | 90.0 | 45.0 | 3.5 | 0.000000 |
| set 33 (cell 19) | 4 | 50 | 90.0 | 45.0 | 4.0 | 0.000000 |
| set 34 (cell 19) | 10 | 50 | 90.0 | 45.0 | 3.5 | 0.000000 |
| set 35 (cell 20) | 10 | 50 | 90.0 | 45.0 | 3.1 | 0.000000 |

Diffusion is measured along the $X$ (long) axis of the fibre. For most cells, two different size regions were bleached per fibre. The width of the bleached region in pixels is indicated for each data set (one pixel is 0.147 µm wide). Intentional bleaching was performed between time points 20 and 21. Two-parameter, $D$ and $β$, fits were performed to long acquisition times, either to time points 21 to 200 (~40 s) or time points 21 to 500 (~110 s). One-parameter fits were also performed to only the initial recovery curve (points 21 to 50, or ~7 s). For the latter fits, bleaching during imaging is too small to fit $β$, so we fix $β = 0$. Results of fits to FRAP curves for GFP and huDysGFP are presented for a model using either the actual fibre length and bleach position or with the standard length of 90 µm and a bleach position at 45 µm. Note that in most GFP cases, the difference between the fitted $D$ values for standard and actual lengths and bleach positions is less than 1 µm²/s. This is smaller than our estimate of the uncertainties in these $D$ values, which is several µm²/s. For huDysGFP, there was no difference within two significant figures. Due to the smaller diffusion constants of huDysGFP, varying the cell length within these limits makes no difference to the values of $D$. During a ~30-s experiment, the bleached profile is not affected by a fibre tip ~45 µm away, for values of $D$ typical of huDysGFP. Values of $D$ in the main text are obtained using all $D$ values obtained for two-parameter fits to data to point 200, using model cells with the standard cell length and bleach position.

hatched squares [unbleached tip] with the respective open or hatched triangles [cytoplasm]). In most cases, at least 50% of the total recovery of unnormalised unbleached minus cytoplasm curves occur within 10 s (*Table 2*). This indicates that there is an unaccounted third pool of huDysGFP, in addition to the immobile-bound pool and the cytoplasmic-free diffusible pool: a rapidly turning-over dynamic bound huDysGFP pool.

Recovery that is presumably due to the bound mobile pool is seen in fibres with no detectable or very low levels of cytoplasmic unbound huDysGFP (*Table 2*). However, the extent of the recovery is still dependent on the amount of cytoplasmic huDysGFP in a fibre (*Figure 6B,F*). Note that in the scatter plot of fractional recovery against cytoplasmic huDysGFP intensity, fibre tips with negligible cytoplasmic huDysGFP show very low recovery (*Figure 6B*). This suggests that cytoplasmic protein is required for the dynamic recovery. Interestingly, tips with high cytoplasmic huDysGFP intensity near the tip do not necessarily recover more (compare *Figure 6C,E,F*) indicating real tip-to-tip variation in the dynamic bound pool. The two tips of the same fibre behave similarly in some but not all cases (*Table 2*).

Most of the recovery that is detected occurs within a few seconds in most cases (*Table 2*). Given that $D_{\text{huDysGFP}}$ is a few µm² s⁻¹, diffusion times across the tip region will be of order a second. Thus, recovery of the dynamic bound pool is fast enough that it may be limited by diffusion. If there is a characteristic binding lifetime, it is at most a few seconds. In summary, our data indicate that most fibres have two pools of huDysGFP bound at their tips, a tightly bound pool stable for at least minutes and another with a sub-second to a few seconds turnover time.

## Lateral immobility of bound Dystrophin

To confirm that the mobility of bound Dystrophin can only be due to exchange with the free-diffusing pool, we tested for lateral mobility of bound huDysGFP, that is, we searched for evidence that huDysGFP can move along the membrane without unbinding and becoming part of the cytoplasmic pool. We examined intensity profiles along the $X_T$ axis, parallel to the tip membrane (*Figure 7B* and

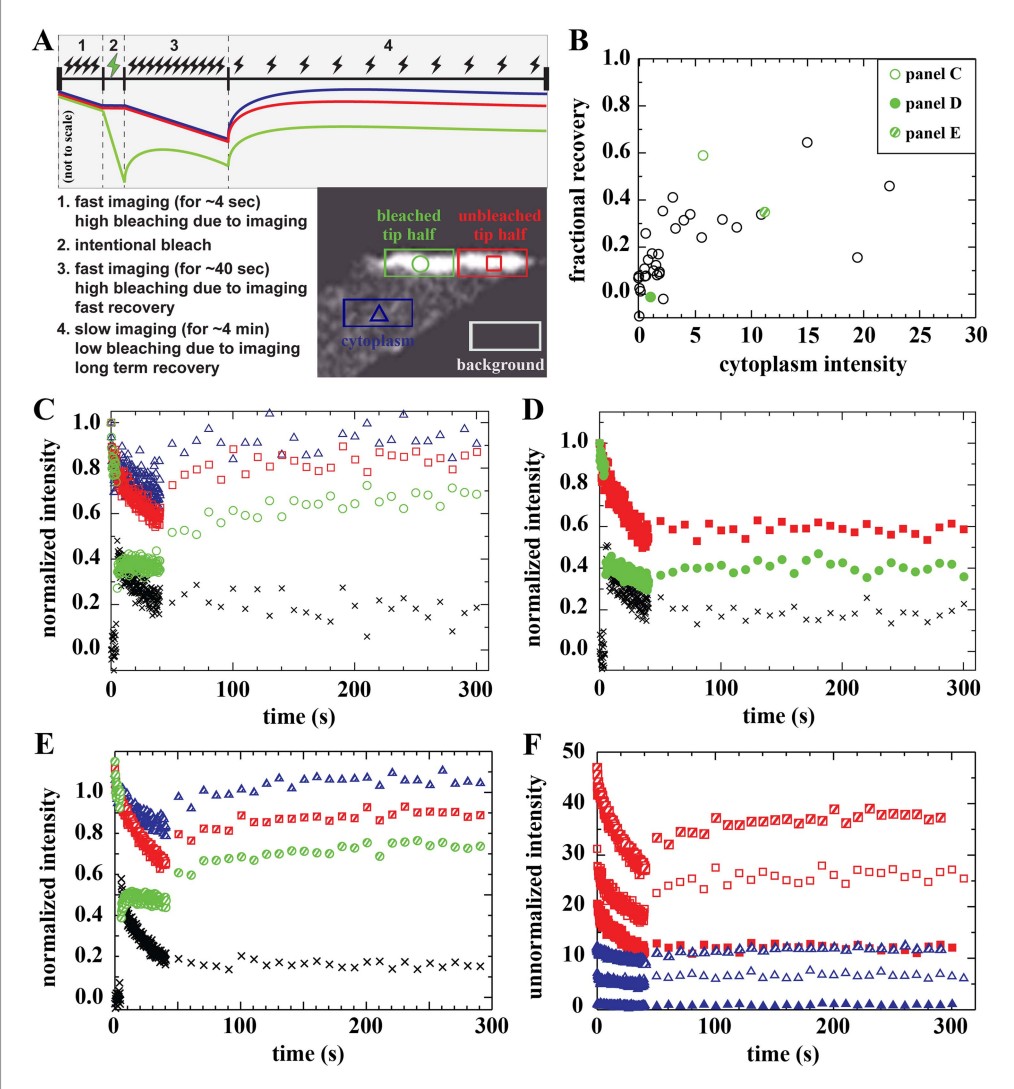

**Figure 6**. Analysis of bound Dystrophin. (**A**) FRAP protocol for tips and schematic FRAP curves showing effect of bleaching-due-to-imaging at high and low imaging frequency. Colour-coding of regions analysed in huDysGFP-expressing cell tips, correspond to traces in **A–E**. (**B**) Scatter plot of fractional recovery in tip pixels as a function of the cytoplasmic intensity. (**C–E**) Examples of normalized fluorescence curves, for tip FRAP recovery of three examples (indicated in **B**) of different tip recovery levels (**C** > **E** > **D**) and cytoplasmic intensity (**E** > **C** > **D**). Intensities in intentionally bleached region (green circles) are lower than in the unbleached tip region (red squares), yielding a difference (black crosses = red − green). Cytoplasm recovers almost fully (blue triangles). See **Table 2** (**C** = tip7, **D** = tip27, **E** = tip32). (**F**) Examples of unnormalized fluorescence curves for unbleached tip (squares) and cytoplasm (triangles), in tips shown in **C** (open symbols), **D** (closed symbols), and **E** (hatched symbols). Note that absolute intensity recovery in tip is larger than in the cytoplasm, but higher cytoplasmic intensity does not result in higher tip recovery (**C** vs **E**).

**Figure 4D**). After deliberately bleaching part of the tip, we looked for a gradient in fluorescence recovery along $X_T$ that could arise from mobility of bound Dystrophin. If the bound Dystrophin is mobile at the tip in the plane of the membrane, bleached bound huDysGFP would leave the deliberately bleached half of the tip and be replaced by fluorescent huDysGFP from the unbleached half. This would help drive fluorescence recovery and would be especially pronounced at the boundary between the bleached and unbleached halves of the tip, generating a gradient of fluorescence along the $X_T$ axis. We see no evidence of such gradients (**Figure 7B**). In particular, in the middle panel of **Figure 7B**, although there is substantial recovery (compare the turquoise crosses with

**Table 2**. Analysis of FRAP data on the tips of huDysGFP-expressing cells in wild-type embryos

| Tip number (cell number) | Cytoplasm intensity | Fractional recovery | Final normalized unbleached-bleached intensities | Unbleached-cytoplasm 50% recovery |
|---|---|---|---|---|
| 1 (cell 1) | 5.80 | 0.24 | 0.14 | <10 s |
| 2 (cell 1) | 4.58 | 0.34 | 0.23 | <10 s |
| 3 (cell 2) | 21.58 | 0.16 | 0.25 | no recovery |
| 4 (cell 2) | 21.90 | 0.46 | −0.06 | <20 s |
| 5 (cell 3) | 8.68 | 0.28 | 0.22 | <10 s |
| 6 (cell 3) | 4.04 | 0.31 | 0.14 | <10 s |
| 7 (cell 4) | 5.94 | 0.59 | 0.06 | <10 s |
| 8 (cell 4) | 15.03 | 0.64 | 0.09 | <10 s |
| 9 (cell 5) | 0.80 | 0.08 | 0.31 | no recovery |
| 10 (cell 5) | 1.81 | 0.17 | 0.36 | <10 s |
| 11 (cell 6) | 0.47 | 0.11 | −0.03 | <10 s |
| 12 (cell 7) | −0.08 | 0.07 | 0.38 | <10 s |
| 13 (cell 7) | 0.02 | 0.08 | 0.17 | <10 s |
| 14 (cell 8) | 0.50 | 0.08 | 0.13 | <10 s |
| 15 (cell 8) | 2.02 | 0.09 | −0.03 | <10 s |
| 16 (cell 9) | 2.22 | −0.02 | 0.10 | <10 s |
| 17 (cell 10) | 0.73 | 0.26 | 0.24 | <30 s |
| 18 (cell 11) | 1.26 | 0.10 | 0.31 | <10 s |
| 19 (cell 12) | 2.31 | 0.35 | 0.07 | <30 s |
| 20 (cell 12) | 1.63 | 0.08 | 0.20 | <10 s |
| 21 (cell 13) | 3.18 | 0.41 | 0.06 | <10 s |
| 22 (cell 14) | 0.07 | 0.02 | 0.24 | <10 s |
| 23 (cell 15) | 0.93 | 0.15 | 0.15 | <10 s |
| 24 (cell 16) | 0.15 | −0.09 | 0.16 | <30 s |
| 25 (cell 17) | 11.29 | 0.34 | 0.26 | >30 s |
| 26 (cell 17) | 7.88 | 0.32 | −0.03 | <20 s |
| 27 (cell 18) | 1.06 | −0.01 | 0.26 | <10 s |
| 28 (cell 19) | 2.05 | 0.09 | 0.22 | <20 s |
| 29 (cell 20) | 0.99 | 0.17 | 0.15 | <10 s |
| 30 (cell 21) | 0.15 | 0.02 | 0.45 | <10 s |
| 31 (cell 22) | 1.52 | 0.12 | 0.28 | <20 s |
| 32 (cell 22) | 11.86 | 0.35 | 0.15 | <10 s |
| 33 (cell 23) | 3.26 | 0.28 | 0.12 | <10 s |

Cytoplasm intensity is the background-subtracted intensity on an 8-bit scale. It is calculated in a rectangular region of a few hundred pixels inside the cell but away from the tip and is averaged over images 4 to 20 (the last one before bleaching). Fractional recovery is the ratio $R_T/I_T$, where, $R_T$ is the intensity recovery in the tip, and $I_T$ is the average pre-bleach intensity in the bleached region. $R_T$ is the average intensity in the bleached region in the final time point (200), minus that in the first point after bleaching (21), and $I_T$ is averaged over images 4 to 20. Final normalised unbleached minus bleached intensities is the difference between the average normalized intensities in the unbleached and bleached regions, derived from the average of the final 20 time points (181–200). Unbleached minus cytoplasm 50% recovery evaluates if at least 50% of the final recovery of unbleached minus cytoplasm curves was rapidly attained, at the first (<10 s), second (<20 s), or later time points after switching from fast to slow acquisition rates. Note that both tips of some fibres were analysed.

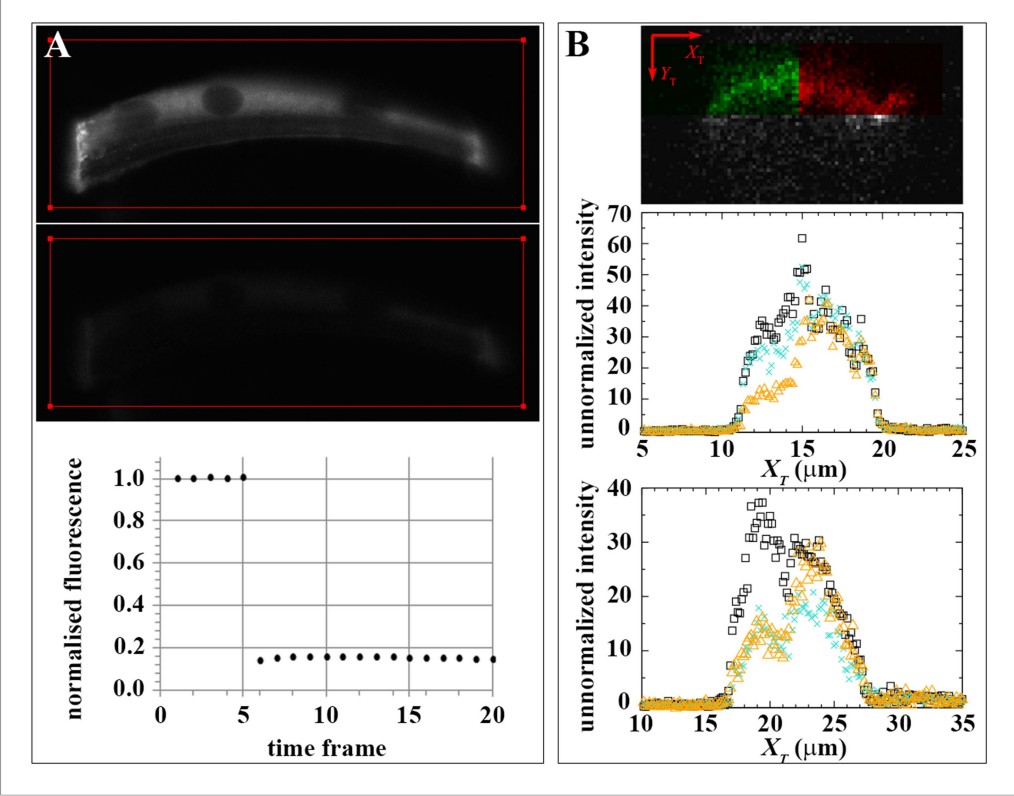

**Figure 7**. Effect of dark state and lateral mobility on huDysGFP intensity recovery. (**A**) To evaluate the recovery fraction due to dark state, huDysGFP was bleached in entire muscle fibres in vivo. Images shown were taken at $t = 0$ (top panel) and right after bleaching (middle panel); the red line defines the bleached region. Lower panel: plot of normalised fluorescence shows very low recovery after photobleaching (0.6%), presumably due to a shift from dark state to excitable huDysGFP. (**B**) FRAP tests for lateral mobility of bound huDysGFP. Top panel: initial image acquired from muscle fibre tip 7 showing the area to be bleached in green and the unbleached tip region in red. Middle and bottom panels: one-dimensional profiles along $X_T$ for an example of substantial recovery (middle panel, tip7) and little recovery (bottom panel, tip27). Profiles are shown at three time points: before deliberate bleaching ($t = 0.93$ s, black squares), just after bleaching ($t = 5.3$ s, orange triangles), and at the end of the experiment ($t = 230$ s, turquoise crosses). The intensity at each point is the average (background corrected) unnormalized intensity over the 20 pixel strip along the $Y_T$ axis and averaged over 3 images, at the given time, plus the previous and next images.

the immediately post-bleach orange triangles), there is no overall tendency in the intensity in the unbleached part of the tip to decrease from right to left as the bleached tip area is approached. The same applies to cases of little recovery (*Figure 7C*), where only substantial bleaching-due-to-imaging is observed on the unbleached tip half. The lack of evidence of lateral mobility of bound huDysGFP argues in favour of the existence of a bound-mobile pool with a fast turnover rate responsible for the fast fluorescence recovery observed in most fibres.

## Human Dystrophin efficiently rescues zebrafish dystrophic embryos and two bound pools are still found in the absence of competition with endogenous Dystrophin

The experiments above were performed in wild-type zebrafish embryos. We asked whether the mobile-bound Dystrophin pool observed may result from competition of the exogenous human Dystrophin with endogenous zebrafish Dystrophin. To address this question, huDysGFP was expressed in Dystrophin-null zebrafish embryos (*dmd*$^{ta222a/ta222a}$).

We first evaluated the ability of human Dystrophin to rescue the zebrafish dystrophic fibres. Typically, in the absence of Dystrophin, the zebrafish muscle fibres detach upon contraction. At 3 days post fertilisation, nearly all *dmd*$^{ta222a/ta222a}$ embryos show signs of dystrophy (*Figure 8A*). To quantify

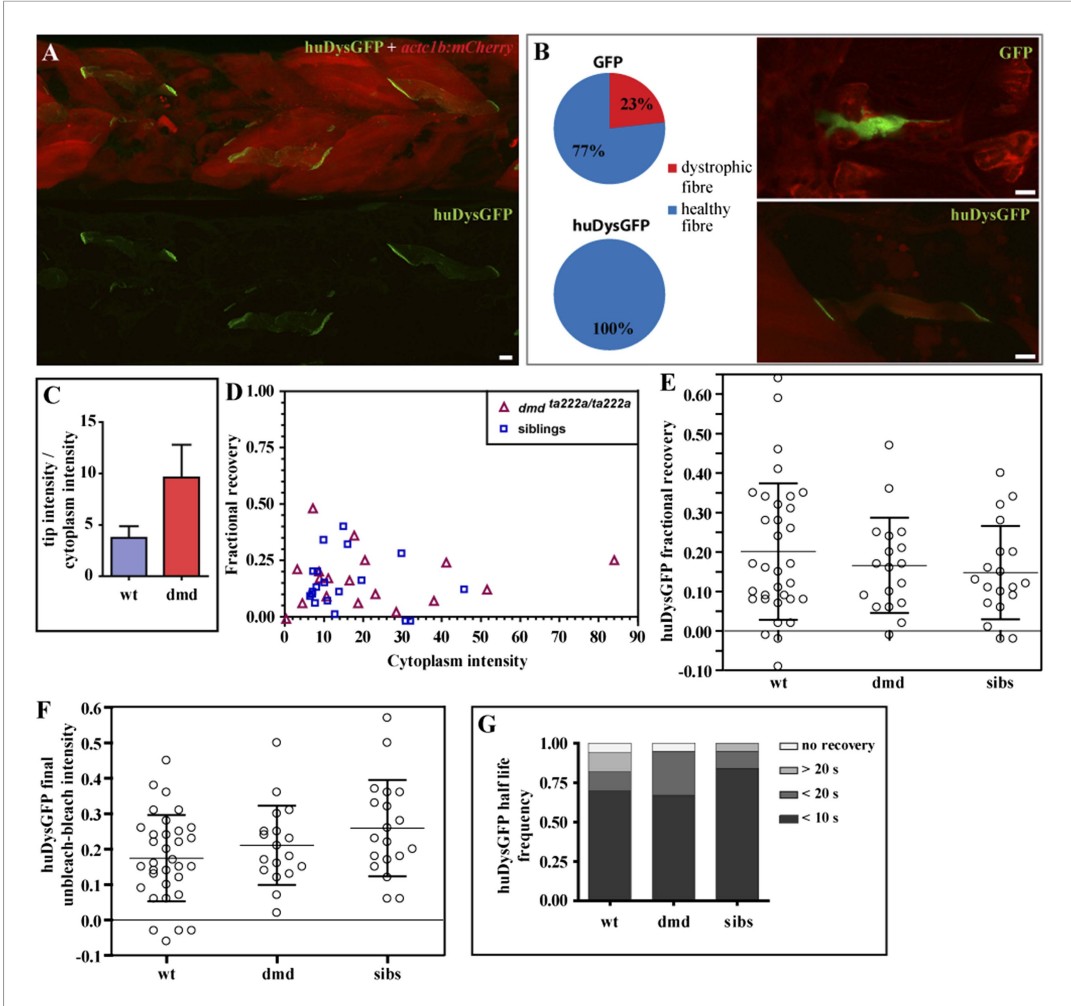

**Figure 8**. huDysGFP rescuing and binding dynamics in *dmd^{ta222a/ta222a}* embryos. (**A**) 3 dpf *dmd^{ta222a/ta222a}* zebrafish embryo with typical dystrophic muscles as shown by *actc1b:mCherry* reporter (red) in vivo, with several healthy fibres expressing huDysGFP (green). (**B**) Control GFP mosaically expressed in *dmd^{ta222a/ta222a}* embryos is found in both healthy (77%) and dystrophic (23%) fibres (N = 56). Expression of huDysGFP fully rescues the dystrophic phenotype in *dmd^{ta222a/ta222a}* muscle fibres, as no cells expressing huDysGFP were found detached or unhealthy in any visible aspect (N = 56). p = 0.000126 in Chi-square test for significance between GFP and huDysGFP. GFP and huDysGFP-positive cells in regions of very dystrophic muscles in *dmd^{ta222a/ta222a}* zebrafish embryos are shown. The *actc1b:mCherry* reporter filling up the cytoplasm and huDysGFP expression at the tip suggest that the fibre structure is kept intact even without support from neighbouring cells, unlike the GFP-positive cell. (**C**) huDysGFP ratio tip intensity/cytoplasm intensity in wild-type (mean = 3.7 ± 1.1 s.e.m.; n = 33) and *dmd^{ta222a/ta222a}* (mean = 9.6 ± 3.2 s.e.m.; n = 13) zebrafish embryos. In the wild-type background, where huDysGFP competes with endogenous Dystrophin for available binding sites, the average ratio is 2.5 times lower than in the mutant background (p = 0.03), where huDysGFP can occupy all available sites. (**D**) Scatter plot of fractional recovery in bleached tip pixels as a function of the cytoplasmic intensity. (**E**) Comparative scatter plots, with mean and SD, of huDysGFP fractional recovery in bleached tip pixels in wild-type (wt), *dmd^{ta222a/ta222a}* (dmd) and their siblings (sibs). There were no statistically significant differences between groups as determined by one-way ANOVA [F(2,67) = 0.8628, ns]. (**F**) Comparative scatter plots, with mean and SD, of huDysGFP final unbleached tip minus bleached tip intensities in wild-type (wt), *dmd^{ta222a/ta222a}* (dmd) and their siblings (sibs). There were no statistically significant differences between groups as determined by one-way ANOVA [F(2,67) = 2.845, ns]. (**G**) huDysGFP fraction of cases showing no recovery, or 50% recovery at the first (<10 s), second (<20 s), or later (>20 s) time points, calculated from unbleached tip minus bleached tip intensities, in wild-type (wt), *dmd^{ta222a/ta222a}* (dmd) and their siblings (sibs). There were no statistically significant differences between groups as determined by one-way ANOVA [F(2,67) = 0.1521, ns]. Scale bars = 10 μm.

rescue efficiency, we first evaluated the percentage of cells that detach by mosaically expressing GFP in $dmd^{ta222a/ta222a}$ embryos. Injecting a GFP control plasmid did not affect muscle fibres in siblings (N = 39), but about 23% of the GFP positive fibres in $dmd^{ta222a/ta222a}$ embryos detached (*Figure 8B*). In marked contrast, cells expressing huDysGFP looked healthy and no detachment was found, suggesting full rescue of the dystrophic phenotype in $dmd^{ta222a/ta222a}$ muscle fibres (*Figure 8B*; p = 0.000126 in Chi-square test for significance between GFP and huDysGFP).

Next, huDysGFP intensity in the whole fibre and locally at the fibre tips was measured in wild-type embryos, where it coexists with the endogenous Dystrophin and in $dmd^{ta222a/ta222a}$ dystrophic embryos (*Figure 8C*). In the absence of competition with the endogenous protein, huDysGFP appears to occupy more of the available binding sites, showing a 2.5-fold increase in the intensity ratio tips: cytoplasm compared to that found in wild-type background. This indicates that fibres expressing the same intensity in the cytoplasm can accumulate more at the fibre tips in the mutant background, consistent with the view that huDysGFP overexpression in the wild-type background does not displace all endogenous zebrafish Dystrophin.

Finally, tip FRAP curves of huDysGFP in $dmd^{ta222a/ta222a}$ and siblings were analysed. Regardless of the genotype, most tips show the three typical signatures of an immobile-bound population described above: only partial recovery in the tip bleached region independently of the cytoplasmic intensity (*Figure 8D,E*; *Table 3*), the normalized intensity difference between the unbleached and bleached regions in most tips reaches a plateau above zero (*Figure 8F*; *Table 3*), and unbleached tip half intensity shows higher drop than the cytoplasm in the fast acquisition phase (see original curves in *Bajanca et al., 2015*). Also regardless of the genotype, the tip intensity registers a rapid recovery that is uncharacteristic of an immobile fraction while higher than the estimated contribution of the cytoplasmic pool. Evidence is clear when comparing events on switching between fast and slow acquisition in the unbleached tip half and cytoplasm unnormalised intensity curves. At least 50% of the total recovery occurs within 10 s in most fibre tips, too fast for the immobile-bound pool (*Figure 8G*; *Table 3*). Overall, our analysis found no significant differences between huDysGFP binding dynamics in wild-type, $dmd^{ta222a/ta222a}$ or their siblings. These results suggest that the presence of two bound populations of huDysGFP with different turnover times is not due to competition with endogenous zebrafish Dystrophin.

## Two bound populations also characterize the dynamics of zebrafish Dystrophin-GFP

We asked whether the labile membrane-bound pool may be a consequence of weaker binding between the human Dystrophin protein and the zebrafish endogenous protein complexes. To address this question, we analysed the dynamics of overexpressed zebrafish Dystrophin: GFP-tagged zebrafish Dystrophin (zfDysGFP). Similar to huDysGFP, the overexpression of zfDysGFP results in mosaic expression and variable levels of accumulation both at the muscle fibre tips and cytoplasm (*Figure 9A*; *Table 4*). The cytoplasmic zfDysGFP diffusion dynamics was analysed by FRAP. $D_{zfDysGFP}$ best-fit values (fitted to data over ~40 s) ranged from 0.6 to 6.7 µm² s⁻¹, with a mean of 2.9 µm² s⁻¹ (*Figure 9B*). zfDysGFP has statistically significantly lower $D$ than GFP (p < 0.001) but shows no difference with huDysGFP (*Figure 9B*).

Next, we analysed zfDysGFP dynamics at the fibre tips, in a wild-type context, where it competes for binding with endogenous Dystrophin, or in $dmd^{ta222a/ta222a}$ mutants (*Figure 9A,C*). Exogenous zfDysGFP is able to rescue the dystrophic phenotype in $dmd^{ta222a/ta222a}$ mutants (32/32, p = 0.003; *Figure 9C*). Like huDysGFP, most zfDysGFP tips show the typical signatures of an immobile-bound population regardless of the genotype (*Figure 9D–F*; *Table 4*). There are no statistically significant differences between the fractional recoveries of Dystrophin of the different species and in the different genetic backgrounds (*Figure 9E*). However, the final unbleached minus bleached intensity of huDysGFP is statistically significantly lower than zfDysGFP (*Figure 9F*). Using a two-way ANOVA to test the effect of genotypic background (wild-type or $dmd^{ta222a/ta222a}$) and Dystrophin origin (human or zebrafish) on the immobile fraction, we observed a significant effect of Dystrophin origin (human vs zebrafish) [F(1,87) = 11.21, p = 0.0012] but not of the host genotype [F(1,87) = 0.02326, p = 0.8791], and there was no significant interaction between origin and genotype [F(1,87) = 1.367, p = 0.2455]. While a final unbleached minus bleached intensity above zero indicates the existence of an immobile-bound fraction, its value is not a direct measurement of the amount of immobile-bound Dystrophin.

**Table 3**. Analysis of FRAP data on the tips of huDysGFP-expressing cells in $dmd^{ta222a/ta222a}$ embryos and siblings

| Tip number | Embryo genotype | Cytoplasm intensity | Fractional recovery | Final normalized unbleached-bleached intensities | Unbleached-cytoplasm 50% recovery |
|---|---|---|---|---|---|
| 1 | $dmd^{ta222a/ta222a}$ | 51.56 | 0.12 | 0.18 | <10 s |
| 2 | $dmd^{ta222a/ta222a}$ | 23.12 | 0.10 | 0.31 | <10 s |
| 3 | $dmd^{ta222a/ta222a}$ | 38.04 | 0.07 | 0.25 | no recovery |
| 4 | $dmd^{ta222a/ta222a}$ | 83.99 | 0.25 | 0.02 | <10 s |
| 5 | $dmd^{ta222a/ta222a}$ | 4.46 | 0.06 | 0.30 | <10 s |
| 6 | $dmd^{ta222a/ta222a}$ | 0.34 | −0.01 | 0.21 | <10 s |
| 7 | $dmd^{ta222a/ta222a}$ | 3.16 | 0.21 | 0.17 | <20 s |
| 8 | $dmd^{ta222a/ta222a}$ | 16.49 | 0.16 | 0.24 | <10 s |
| 9 | $dmd^{ta222a/ta222a}$ | 18.67 | 0.06 | 0.07 | <10 s |
| 10 | $dmd^{ta222a/ta222a}$ | 8.73 | 0.20 | 0.25 | <10 s |
| 11 | $dmd^{ta222a/ta222a}$ | 7.12 | 0.47 | 0.12 | <20 s |
| 12 | $dmd^{ta222a/ta222a}$ | 11.06 | 0.17 | 0.15 | <20 s |
| 13 | $dmd^{ta222a/ta222a}$ | 10.65 | 0.09 | 0.23 | <10 s |
| 14 | $dmd^{ta222a/ta222a}$ | 28.41 | 0.02 | 0.13 | <20 s |
| 15 | $dmd^{ta222a/ta222a}$ | 8.91 | 0.17 | 0.14 | <10 s |
| 16 | $dmd^{ta222a/ta222a}$ | 41.16 | 0.24 | 0.16 | <10 s |
| 17 | $dmd^{ta222a/ta222a}$ | 17.71 | 0.36 | 0.50 | <20 s |
| 18 | $dmd^{ta222a/ta222a}$ | 20.43 | 0.25 | 0.36 | <10 s |
| 19 | sibling | 30.85 | −0.02 | 0.20 | <10 s |
| 20 | sibling | 45.81 | 0.12 | 0.17 | <30 s |
| 21 | sibling | 13.89 | 0.11 | 0.23 | <10 s |
| 22 | sibling | 7.05 | 0.10 | 0.33 | <10 s |
| 23 | sibling | 12.78 | 0.01 | 0.32 | <10 s |
| 24 | sibling | 6.48 | 0.09 | 0.26 | <10 s |
| 25 | sibling | 10.95 | 0.07 | 0.28 | <10 s |
| 26 | sibling | 8.35 | 0.20 | 0.22 | <10 s |
| 27 | sibling | 10.14 | 0.15 | 0.15 | <10 s |
| 28 | sibling | 7.15 | 0.11 | 0.50 | <10 s |
| 29 | sibling | 7.77 | 0.06 | 0.57 | <10 s |
| 30 | sibling | 19.64 | 0.16 | 0.18 | <10 s |
| 31 | sibling | 16.08 | 0.32 | 0.36 | <10 s |
| 32 | sibling | 8.11 | 0.13 | 0.37 | <10 s |
| 33 | sibling | 7.30 | 0.20 | 0.36 | <10 s |
| 34 | sibling | 14.99 | 0.40 | 0.06 | <20 s |
| 35 | sibling | 29.82 | 0.28 | 0.06 | <20 s |
| 36 | sibling | 9.95 | 0.34 | 0.18 | <10 s |
| 37 | sibling | 32.10 | −0.02 | 0.12 | <10 s |

See **Table 2** for detailed information.

However, we cannot exclude the possibility that huDysGFP forms less, or less stable, immobile-bound links than zfDysGFP, in the zebrafish environment. The background genotype appears not to influence this trait, suggesting that if there is a difference in the immobile fraction that is independent from competition with endogenous Dystrophin.

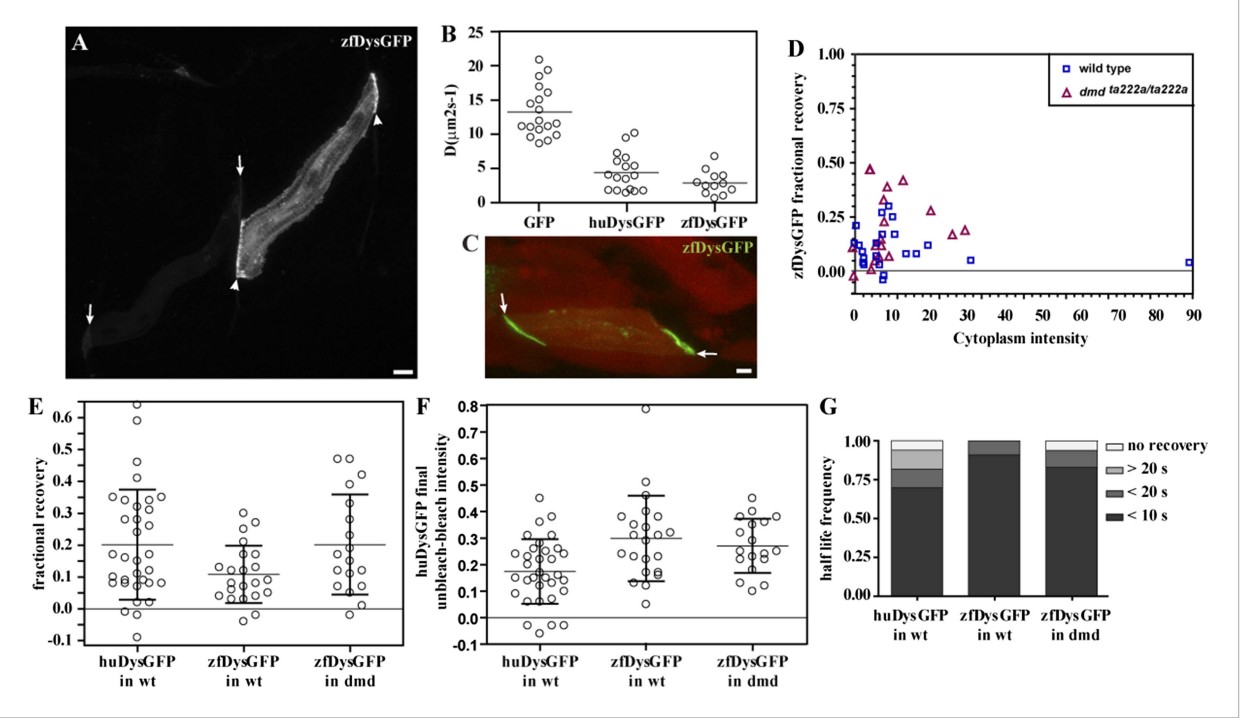

**Figure 9**. zfDysGFP dynamics in wild-type and *dmd*[*ta222a/ta222a*] embryos. (**A**) zfDysGFP variable intensity of expression in muscle fibres of 2 dpf wild-type embryo. Arrows point to low and arrowheads to high expressing tips. (**B**) Comparative scatter plots of $D_{GFP}$, $D_{huDysGFP}$ and $D_{zfDysGFP}$. One-way ANOVA revealed a statistically significant difference between groups [$F(2,44) = 57.08$, $p < 0.0001$]. Tukey post-hoc test revealed that $D_{huDysGFP}$ ($4.4 \pm 2.7$ µms²s⁻¹) and $D_{zfDysGFP}$ ($2.9 \pm 1.7$ µms²s⁻¹) are not statistically significant but are statistically significantly lower ($p < 0.001$) than $D_{GFP}$ ($13.2 \pm 3.7$ µms²s⁻¹). (**C**) A rescued zfDysGFP (green) fibre within a 2 dpf *dmd*[*ta222a/ta222a*] zebrafish embryo with otherwise typical dystrophic muscles as shown by *actc1b:mCherry* (red, note extensive gaps in muscle) reporter in vivo. (**D**) Scatter plot of fractional recovery in bleached tip pixels as a function of the cytoplasmic intensity. (**E**) Comparative scatter plots, with mean and SD, of the fractional recovery in bleached tip pixels of huDysGFP in wild-type (wt) embryos, and zfDysGFP in wild-type and *dmd*[*ta222a/ta222a*] (dmd) embryos. There were no statistically significant differences between groups as determined by one-way ANOVA [$F(2,70) = 3.019$, ns]. (**F**) Comparative scatter plots, with mean and SD, of final unbleached tip minus bleached tip intensities of huDysGFP in wild-type embryos, and zfDysGFP in wild-type and *dmd*[*ta222a/ta222a*] embryos. One-way ANOVA revealed a statistically significant difference between groups [$F(2,70) = 6.818$, $p = 0.002$]. Tukey post-hoc test revealed that zfDysGFP in wild-type ($0.3 \pm 0.16$, $n = 22$) and *dmd*[*ta222a/ta222a*] embryos ($0.3 \pm 0.1$, $n = 18$) are not statistically significant but huDysGFP ($0.17 \pm 0.12$, $n = 33$) is statistically significantly lower than zfDysGFP in wild-type ($p < 0.01$) and in *dmd*[*ta222a/ta222a*] embryos ($p < 0.05$). (**G**) Fraction of cases showing no recovery, or 50% recovery at the first (<10 s), second (<20 s), or later (>20 s) time points, calculated from unbleached tip minus bleached tip intensities, in huDysGFP in wild-type embryos, and zfDysGFP in wild-type and *dmd*[*ta222a/ta222a*] embryos. There were no statistically significant differences between groups as determined by one-way ANOVA [$F(2,70) = 1.405$, ns]. Scale bars = 10 µm.

Like huDysGFP, most zfDysGFP-expressing fibres show a fast recovery phase that is higher than the estimated contribution of the cytoplasmic pool. Typically, at least 50% of the total recovery of unbleached tip half minus cytoplasm intensity curves occurs within 10 s in the majority of the cases, regardless of species and genotype (*Figure 9G*; *Table 4*). This recovery is very fast and the characteristic immobile pool plateau is soon reached, suggesting the existence of an additional bound pool of zfDysGFP with rapid turnover. These results show that, like huDysGFP, zfDysGFP can be found in three populations, one diffusible and two bound with different binding lifetimes. Therefore, regardless of any differences between huDysGFP and zfDysGFP, our results confirm that the mobile-bound pool previously found for the human Dystrophin is not caused by weaker interactions with a different species environment.

## Endogenous zebrafish Dystrophin diffusion and binding dynamics

Analysis of FRAP curves of both huDysGFP and zfDysGFP indicated that when a low level of diffusible Dystrophin is detected in cytoplasm, there can be a significant amount of bound-mobile pool. However, there is the possibility that the labile-bound pool is caused by an excess of cytoplasmic

**Table 4**. Analysis of FRAP data on the tips of zfDysGFP-expressing cells in wild-type and $dmd^{ta222a/ta222a}$ embryos

| Tip number (cell number) | Embryo genotype | Cytoplasm intensity | Fractional recovery | Final normalized unbleached-bleached intensities | Unbleached-cytoplasm 50% recovery |
|---|---|---|---|---|---|
| 1 (cell 1) | wild type | 14.59 | 0.08 | 0.46 | <10 s |
| 2 (cell 2) | wild type | 5.06 | 0.07 | 0.24 | <10 s |
| 3 (cell 3) | wild type | 9.44 | 0.17 | 0.12 | <20 s |
| 4 (cell 4) | wild type | 6.51 | 0.17 | 0.38 | <10 s |
| 5 (cell 4) | wild type | 8.02 | 0.30 | 0.05 | <10 s |
| 6 (cell 5) | wild type | 79.28 | 0.04 | 0.22 | <10 s |
| 7 (cell 6) | wild type | 1.78 | 0.09 | 0.51 | <20 s |
| 8 (cell 6) | wild type | 1.01 | 0.12 | 0.38 | <10 s |
| 9 (cell 7) | wild type | 0.28 | 0.21 | 0.13 | <10 s |
| 10 (cell 8) | wild type | 27.46 | 0.05 | 0.16 | <10 s |
| 11 (cell 9) | wild type | 12.13 | 0.08 | 0.17 | <10 s |
| 12 (cell 9) | wild type | 17.34 | 0.12 | 0.40 | <10 s |
| 13 (cell 10) | wild type | 5.09 | 0.13 | 0.23 | <10 s |
| 14 (cell 10) | wild type | 2.14 | 0.03 | 0.34 | <10 s |
| 15 (cell 11) | wild type | −0.13 | 0.13 | 0.80 | <10 s |
| 16 (cell 12) | wild type | 2.09 | 0.06 | 0.17 | <10 s |
| 17 (cell 12) | wild type | 1.94 | 0.04 | 0.31 | <10 s |
| 18 (cell 13) | wild type | 5.80 | 0.03 | 0.29 | <10 s |
| 19 (cell 13) | wild type | 6.65 | −0.04 | 0.32 | <10 s |
| 20 (cell 14) | wild type | 6.42 | 0.27 | 0.31 | <10 s |
| 21 (cell 14) | wild type | 6.87 | −0.02 | 0.35 | <10 s |
| 22 (cell 15) | wild type | 8.96 | 0.25 | 0.23 | <10 s |
| 23 (cell 16) | $dmd^{ta222a/ta222a}$ | 11.40 | 0.42 | 0.25 | <10 s |
| 24 (cell 17) | $dmd^{ta222a/ta222a}$ | 4.70 | 0.05 | 0.35 | <10 s |
| 25 (cell 17) | $dmd^{ta222a/ta222a}$ | 3.81 | 0.01 | 0.40 | <10 s |
| 26 (cell 18) | $dmd^{ta222a/ta222a}$ | 6.77 | 0.33 | 0.12 | <10 s |
| 27 (cell 19) | $dmd^{ta222a/ta222a}$ | −0.38 | −0.02 | 0.36 | <10 s |
| 28 (cell 19) | $dmd^{ta222a/ta222a}$ | −0.60 | 0.11 | 0.32 | <10 s |
| 29 (cell 20) | $dmd^{ta222a/ta222a}$ | 4.86 | 0.12 | 0.38 | <10 s |
| 30 (cell 21) | $dmd^{ta222a/ta222a}$ | 3.45 | 0.47 | 0.24 | <10 s |
| 31 (cell 21) | $dmd^{ta222a/ta222a}$ | 3.60 | 0.47 | 0.25 | <10 s |
| 32 (cell 22) | $dmd^{ta222a/ta222a}$ | 7.60 | 0.39 | 0.10 | <10 s |
| 33 (cell 23) | $dmd^{ta222a/ta222a}$ | 26.07 | 0.19 | 0.18 | <10 s |
| 34 (cell 24) | $dmd^{ta222a/ta222a}$ | 6.08 | 0.15 | 0.45 | <10 s |
| 35 (cell 24) | $dmd^{ta222a/ta222a}$ | 8.03 | 0.07 | 0.38 | no recovery |
| 36 (cell 25) | $dmd^{ta222a/ta222a}$ | 17.93 | 0.28 | 0.22 | <10 s |
| 37 (cell 26) | $dmd^{ta222a/ta222a}$ | 6.21 | 0.12 | 0.23 | <10 s |
| 38 (cell 27) | $dmd^{ta222a/ta222a}$ | 5.62 | 0.07 | 0.13 | <20 s |
| 39 (cell 28) | $dmd^{ta222a/ta222a}$ | 6.88 | 0.23 | 0.22 | <10 s |
| 40 (cell 29) | $dmd^{ta222a/ta222a}$ | 23.05 | 0.17 | 0.29 | <20 s |

See **Table 2** for detailed information.

Dystrophin resulting from overexpression. We questioned whether a mobile-bound pool can also be found in endogenous zebrafish Dystrophin, where a cytoplasmic pool of the protein is not known. We have analysed *Gt(dmd-Citrine)^{ct90a}* zebrafish embryos, in which Citrine was inserted by gene-trap into the endogenous Dystrophin locus, creating fluorescently tagged zfDys (*Trinh et al., 2011*; *Ruf-Zamojski et al., 2015*).

We first searched for signs of cytoplasmic Dystrophin. As every muscle fibre expresses Citrine-tagged Dystrophin (zfDysCitrine; *Figure 10A*), and different tissues may have different background fluorescence, it is not easy to evaluate with confidence whether there are low levels of cytoplasmic zfDysCitrine above the background. We hypothesized that if there is zfDysCitrine in the cytoplasm, we should register a recovery after photobleaching. Control siblings lacking zfDysCitrine show a flat and noisy FRAP curve that reflects the background fluorescence (*Figure 10B*, plot a). In contrast, the very low zfDysCitrine intensity detected in the cytoplasm could be bleached to even lower levels, and a recovery curve is registered, indicating the presence of diffusible zfDysCitrine in the cytoplasm (*Figure 10B*, plot b). The switch from a dark state of Citrine was evaluated and contributes little to the recovery (*Figure 10B*, plot c). Another signature of the presence of diffusible zfDysCitrine was the inverted bell-shaped, instead of top-hat, Gaussian curve that results from diffusion during intentional bleaching (*Figure 10C*). Finally, zfDysCitrine recovery curves can be fit using a diffusion model (*Figure 10D*). $D_{zfDysCitrine}$ best-fit values ranged from 0.9 to 4.3 $\mu m^2\ s^{-1}$ and mean 2.2 $\mu m^2\ s^{-1}$ (*Figure 10E*). There is no significant statistical difference between $D_{zfDysCitrine}$ and $D_{zfDysGFP}$.

To be able to analyse single fibres, some embryos were subjected to heat shock, which causes mosaic disruption of the somites (*Ruf-Zamojski et al., 2015*). Embryos not subjected to heat shock, at different developmental stages (30 hpf, 40 hpf and 48 hpf) were also analysed. However, no trend was obvious when analysing embryos of different developmental stages (*Figure 10F*; *Table 5*). Most tips showed the typical signatures of an immobile-bound pool, as expected, and no statistically significant differences were found between zfDysGFP and zfDysCitrine (*Figure 10F–H*; *Table 5*). There is fractional recovery in most bleached tips that tends to be low (*Figure 10F*, *Table 5*), which in accordance with the results obtained for huDysGFP and zfDysGFP cases when the cytoplasm intensity is just above detectable (*Figures 6B, 8D, 9D*; *Tables 2–4*). Therefore, we found no indications that endogenous zfDysCitrine behave differently from exogenous zfDysGFP regarding their immobile pool signatures.

Despite the low cytoplasmic intensity, many zfDysCitrine tips still show a very fast recovery. At least 50% of the total recovery of the unbleached tip half minus the cytoplasm happens in almost half of the cases within 10 s, and still very fast at less than 20 s in the majority of the remaining cases (*Figure 10I*; *Table 5*). This suggests the presence of a mobile-bound pool, regardless of the statistically significant difference in the time it takes for zfDysCitrine and zfDysGFP to recover (p = 0.0016). The tendency to take longer to recover is likely to reflect the dependency of the mobile-bound pool on cytoplasmic availability of free Dystrophin. This is in accordance with our previous observations that recovery of the mobile-bound pool is fast enough that it may be limited by diffusion. The data therefore indicate that endogenous Dystrophin with an inserted Citrine, like exogenous zfDysGFP and huDysGFP, can be found in two states, immobile and mobile, regardless of a very low, but detectable, cytoplasmic pool.

## Discussion

Altogether, our results show that Dystrophin can exist in three populations in muscle cells in vivo: one freely diffusing in the cytoplasm, a second stably anchored at the plasma membrane that shows no lateral diffusion, and a third, dynamically bound to the plasma membrane. A stable-bound pool is consistent with Dystrophin's well-described function of sustaining mechanical stability upon muscle contraction (*Ibraghimov-Beskrovnaya et al., 1992*; *Levine et al., 1992*; *Ervasti and Campbell, 1993*; *Rybakova et al., 1996*, *2000*). Additionally, the presence of a dynamic pool with weak and short-term binding points to a new aspect of Dystrophin biology. Based on these results, we propose a model for Dystrophin association with the membrane (*Figure 11*). Even very low levels of freely diffusible cytoplasmic Dystrophin can stochastically bind to molecules at the plasma membrane, entering a mobile-bound state (*Figure 11*, arrows). This may represent an intermediate state towards Dystrophin stabilization, possibly more abundant in the immature fibres of the developing embryo (*Figure 11*, dashed arrow 1). Alternatively, Dystrophin may coexist in two distinct bound forms with specific roles, possibly reflecting the mature state of the membrane association.

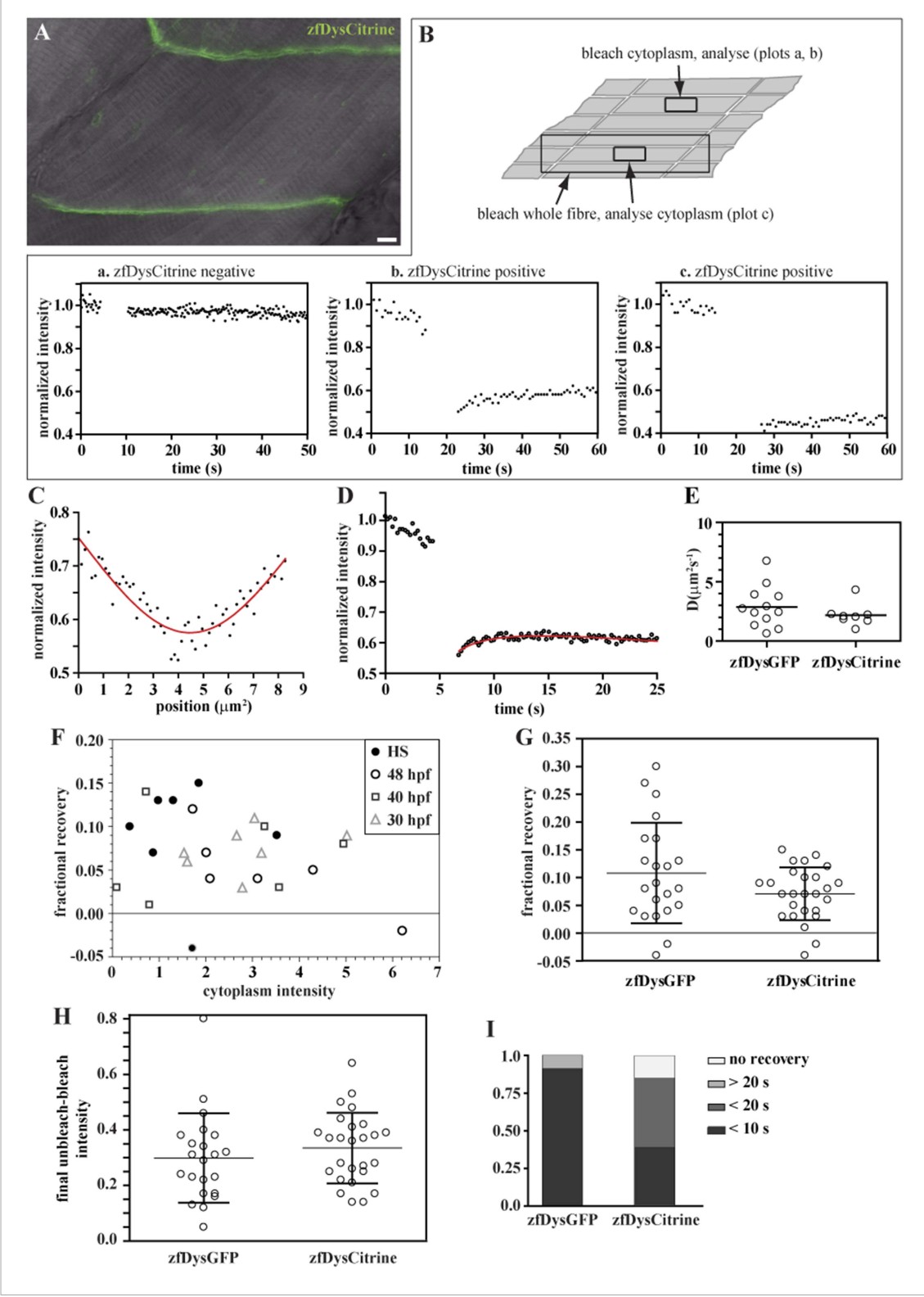

**Figure 10**. Endogenously driven zfDysCitrine dynamics. (**A**) In vivo zfDysCitrine (green) expression in muscle fibres of 2 dpf *Gt(dmd-citrine)*[ct90a] embryo contrasted with transmitted light. (**B**) Schematics of zebrafish muscle fibres showing origin of scatter plots a, b, and c. Bleaching a region in the cytoplasm of Citrine-negative siblings (a) results in a flat curve of background fluorescence intensity, while the same experiment in Citrine-positive embryos (b) results in a significant drop in fluorescence followed by recovery. Bleaching a large region to include the entire fibre abolishes recovery (c), indicating that

*Figure 10. continued on next page*

*Figure 10. Continued*

recovery from a citrine dark state makes a negligible contribution to recovery in (b). (**C**, **D**) Normalized FRAP experimental data and fitting curves of a zfDysCitrine fibre cytoplasm. (**C**) Normalized intensity profile along the *X*-axis at the first time point after bleaching (dots) and Gaussian fit (red line). (**D**) Recovery curves along *X*-axis (dots) and fit of the diffusion model to the post-bleach (red line). (**E**) Comparative scatter plots of $D_{zfDysGFP}$ and $D_{zfDysCitrine}$. *t* test shows no statistically significant differences. (**F**) Scatter plot of fractional recovery in bleached tip pixels as a function of the cytoplasmic intensity, for zfDysCitrine embryos of different developmental stages. HS = heat-shocked embryos. (**G**) Comparative scatter plots, with mean and SD, of the fractional recovery in bleached tip pixels of zfDysGFP and zfDysCitrine. *t* test shows no statistically significant difference. (**H**) Comparative scatter plots, with mean and SD, of final unbleached tip minus bleached tip intensities of zfDysGFP and zfDysCitrine. *t* test shows no statistically significant differences. (**I**) Fraction of cases showing no recovery, or 50% recovery at the first (<10 s), second (<20 s), or later (>20 s) time points, calculated from unbleached tip minus bleached tip intensities, in zfDysGFP and zfDysCitrine. t test shows a statistically significant difference (p = 0.0016). Scale bar = 10 µm.

## Validating Dystrophin over-expression in zebrafish as a model system to study Dystrophin dynamics in vivo

In various fields, transgenic and humanized animal models are a valuable resource where non-invasive methods to study human biology are lacking (*Boverhof et al., 2011*; *Attfield and Dendrou, 2012*; *Akkina, 2013*). Here, we show that exogenous zebrafish and human Dystrophin have subcellular localizations, at both mRNA and protein levels, equivalent to that of endogenous Dystrophin (*Ruf-Zamojski et al., 2015*). Furthermore, exogenous zebrafish and human Dystrophin have diffusion and binding dynamics similar to those of endogenous zebrafish Dystrophin, in spite of the artificially raised cytoplasmic levels caused by over-expression. Furthermore, we found that Dystrophin dynamics is not affected by the position of the fluorescent tag (internally close to the actin binding site [Citrine] or in C-terminal position [GFP]). Importantly, both zebrafish and human Dystrophins were able to rescue the dystrophic phenotype of *dmd*^ta222a/ta222a embryos. Taken together, these data indicate that the zebrafish embryo is a good model system to study the dynamics of human Dystrophin in live muscle cells in vivo using fluorescently tagged versions of the protein. It is important to keep in mind that human Dystrophin may behave differently in zebrafish than in human muscle. The FRAP analysis methodology developed in this study could be applied for studies on human primary muscle cell cultures, or even pluripotent human stem cells differentiated into muscle fibres (*Chal et al., 2015*). However, until a suitable 3D ex-vivo physiologically relevant human muscle system is readily available for routine experimentation, our methodology and findings provide a baseline for future comparative studies. For instance, the strategy presented here can be used to study the effects of shortening the protein on Dystrophin dynamics, as occurs in patients with BMD and planned exon-skipping gene therapies (*Koenig et al., 1989*; *Cirak et al., 2011*; *Konieczny et al., 2013*; *Verhaart et al., 2014*).

## Cytoplasmic Dystrophin

We show that, in all three experimental conditions used (exogenous zfDysGFP and huDysGFP or endogenously-driven zfDysCitrine), part of Dystrophin is found in a cytoplasmic freely diffusing pool. Despite considerable apparent variation in measured diffusion constant (*D*) from fibre to fibre (1.4–10.1 µm² s⁻¹ for $D_{huDysGFP}$, 0.6 to 6.7 µm² s⁻¹ for $D_{zfDysGFP}$, 0.9 to 4.3 µm² s⁻¹ for $D_{zfDysCitrine}$, mean 4 µm² s⁻¹, 3 µm² s⁻¹, and 2 µm² s⁻¹, respectively), it is clear that the mobility of Dystrophin is on average around a fourth that of GFP, at approximately 13 µm² s⁻¹. However, whereas GFP is a small (3 × 3 × 4 nm) globular protein, Dystrophin is thought to be a rod perhaps 100 nm long (*Pons et al., 1990*; *Arrio-Dupont et al., 2000*; *Bhasin et al., 2005*; *Kameta et al., 2010*; *Kinsey et al., 2011*), so a diffusion constant only a quarter that of GFP is surprising. As Dystrophin appears to diffuse, perhaps so-called 'active diffusion' due to energy-using cellular processes (e.g., molecular motors) enhances its apparent mobility in a non-directed manner (*Brangwynne et al., 2008*, *2009*; *Weber et al., 2012*). As whole Dystrophin structure has not been reported, one can imagine that a compact rapidly diffusing Dystrophin conformation may account for Dystrophin dynamics in vivo. For now, it is not known whether cytoplasmic Dystrophin also exists in low amounts in adult human skeletal muscle cells. However, many studies have shown that, in human embryos and foetuses, Dystrophin first appears in the cytoplasm (*Wessels et al., 1991*; *Clerk et al., 1992*; *Chevron et al., 1994*; *Mora et al., 1996*; *Torelli et al., 1999*). Interestingly, a cytoplasmic Dystrophin pool was also found in the adult heart

**Table 5.** Analysis of FRAP data on the tips of ZfDysCitrine expressing cells in *Gt(dmd-Citrine)$^{ct90a}$* embryos

| Tip number | Set | Cytoplasm intensity | Fractional recovery | Final normalized unbleached-bleached intensities | Unbleached-cytoplasm 50% recovery |
|---|---|---|---|---|---|
| 1 | heat shock | 1.71 | −0.04 | 0.37 | <20 s |
| 2 | heat shock | 1.85 | 0.15 | 0.14 | <20 s |
| 3 | heat shock | 0.37 | 0.10 | 0.39 | <20 s |
| 4 | heat shock | 3.52 | 0.09 | 0.14 | <30 s |
| 5 | heat shock | 0.98 | 0.13 | 0.17 | <20 s |
| 6 | heat shock | 0.87 | 0.07 | 0.21 | <20 s |
| 7 | heat shock | 1.30 | 0.13 | 0.17 | <10 s |
| 8 | 48 hpf | 4.30 | 0.05 | 0.36 | <10 s |
| 9 | 48 hpf | 3.11 | 0.04 | 0.37 | <10 s |
| 10 | 48 hpf | 1.72 | 0.12 | 0.53 | <20 s |
| 11 | 48 hpf | 2.09 | 0.04 | 0.39 | <20 s |
| 12 | 48 hpf | 2.01 | 0.07 | 0.42 | <10 s |
| 13 | 48 hpf | 6.21 | −0.02 | 0.41 | <10 s |
| 14 | 40 hpf | 3.26 | 0.10 | 0.27 | <20 s |
| 15 | 40 hpf | 0.72 | 0.14 | 0.25 | <30 s |
| 16 | 40 hpf | 3.57 | 0.03 | 0.26 | <20 s |
| 17 | 40 hpf | 4.95 | 0.08 | 0.22 | <20 s |
| 18 | 40 hpf | 0.79 | 0.01 | 0.25 | <10 s |
| 19 | 40 hpf | 0.09 | 0.03 | 0.28 | <20 s |
| 20 | 30 hpf | 1.53 | 0.07 | 0.38 | <10 s |
| 21 | 30 hpf | 1.60 | 0.06 | 0.48 | <60 s |
| 22 | 30 hpf | 5.02 | 0.09 | 0.50 | <20 s |
| 23 | 30 hpf | 2.66 | 0.09 | 0.64 | <40 s |
| 24 | 30 hpf | 3.04 | 0.11 | 0.44 | <10 s |
| 25 | 30 hpf | 3.19 | 0.07 | 0.37 | <10 s |
| 26 | 30 hpf | 2.78 | 0.03 | 0.28 | <10 s |

In embryos subjected to heat shock (tips 1–7), individual fibres could be selected for FRAP, at 48 hpf, and background levels are taken into account, contrary to the remaining cases (tips 8–26). See **Table 2** for detailed information.

(*Peri et al., 1994*) and in both regenerating fibres (*Kääriäinen et al., 2000*) and differentiating human primary muscle cultures (*Miranda et al., 1988*), where Dystrophin localisation was suggested to recapitulate the embryonic process of Dystrophin deposition, accumulating initially in the cytoplasm and muscle–tendon junctions at the fibre ends, prior to maturing towards costameric localisation. Therefore, it would be particularly interesting to analyse Dystrophin dynamics in the cytoplasm of human dystrophic muscle cells, which undergo repeated cycles of regeneration.

## Immobile- and mobile-bound Dystrophin

Our direct semi-quantitative FRAP analysis of bleaching and recovery allowed identification of two populations of immobile- and mobile-bound Dystrophin. The dynamic mobile-bound pool is found in fibres with undetectably low levels of cytoplasmic huDysGFP, zfDysGFP, or endogenously-driven zfDysCitrine, indicating that the mobile-bound state occurs independently of over-expression or high level cytoplasmic accumulation.

A proportion of the Dystrophin located at muscle fibre tips is in free exchange with the cytoplasmic diffusible pool, whereas a further portion binds stably at muscle fibre tips. Whether this immobile pool

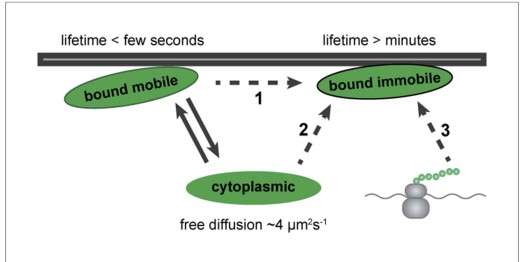

**Figure 11**. Model for Dystrophin membrane association. Dystrophin is present in three states: cytoplasmic, bound mobile, and bound immobile. Switching between cytoplasmic and bound mobile (solid arrows) occurs at a rate of under a few seconds, limited by the diffusion rate. Immobile Dystrophin is stably bound for at least several minutes. Dashed arrows represent three possible routes to stable Dystrophin complex formation: (1) from a bound mobile Dystrophin intermediate, (2) by direct addition from the cytoplasmic pool, (3) by anchoring of nascent Dystrophin polypeptide chains from localized mRNA. These possibilities are not mutually exclusive.

is generated by stabilisation of the mobile-bound pool, or by another route, remains to be determined (*Figure 11*). It is not clear whether the dynamics of early tip-localised Dystrophin is different from the later costameric Dystrophin. The tip region of zebrafish muscle fibres corresponds to myotendinous junction at the ends of human muscle fibres, where Dystrophin is also found enriched in humans (*Zhao et al., 1992*) and in several other mammals, namely mouse (*Samitt and Bonilla, 1990*), rat (*Kääriäinen et al., 2000*), and guinea pig (*Masuda et al., 1992*).

The immobile Dystrophin pool at the cell tips is bound tightly enough to transmit significant (10 pN/molecule) forces for significant (1 s) times, enough to unfold Dystrophin's spectrin domains (*Bhasin et al., 2005*). Dystrophin in the bound mobile pool presumably cannot transmit significant force. However, weak binding would allow response to weak short-lived (sub-pN/molecule, sub-second) forces or may fulfil another function, such as structure assembly, sensing, or signalling. Interestingly, studies of mutants of ezrin, which, like Dystrophin, binds both β-dystroglycan and actin, revealed immobile and mobile membrane bound forms (*Coscoy et al., 2002*; *Spence et al., 2004*). The dynamic membrane-bound ezrin was suggested to be an intermediate conformation state leading to actin anchoring and full complex assembly. Similarly, it is possible that Dystrophin binding to β-dystroglycan facilitates a conformation change to promote actin binding and stabilization of the complex (*Friedel et al., 2006*) (*Figure 11*). Membrane localization, turnover, and clustering of other adhesion molecules such as cadherins is known to be influenced by the tension experienced by the cells (*Delanoë-Ayari et al., 2004*; *Yonemura et al., 2010*; *De Beco et al., 2015*). Thus, it would be interesting to investigate whether muscle contraction might favour the conversion of some of the mobile-bound or cytoplasmic Dystrophin into immobile Dystrophin.

## mRNA accumulation at the tip of muscle cells

Our expression vectors contain a CMV promoter that drives human and zebrafish Dystrophin expression throughout the fish. Nonetheless, we observed that muscle cells accumulate Dystrophin protein much more frequently than other cell types, suggesting that human Dystrophin stabilization is tissue dependent. The availability of suitable docking sites or specific partner proteins may play a role in stabilization (*Le Rumeur et al., 1804*). In addition, sub-cellular accumulation of the mRNA itself may contribute to Dystrophin positioning at the tip. Like zebrafish Dystrophin mRNA, RNA encoding human Dystrophin localized near the tips of fibres. The expression constructs engineered in the present study do not contain the Dystrophin 5′- or 3′-UTR or introns. Instead, the coding sequence is preceded by a standard chimaeric intron. While previous studies showed a role for 5′ and 3′ UTR regions into controlling tissue-specific expression and transcriptional regulation of Dystrophin (see *Larsen and Howard, 2014* and references therein), our results show that UTR regions are dispensable for accumulation of the mRNA at the tip. It is therefore unlikely that Dystrophin RNA accumulation at the tips is due to specific RNA transport since that would most likely require the presence of the untranslated regions (for review see *Kloc et al., 2002*; *Holt and Bullock, 2009*). Thus, the signals controlling the correct localization of the mRNA remain unclear. One possibility is anchorage by numerous nascent protein chains (*Figure 11*). This might then facilitate transition of Dystrophin into the strongly bound form.

## Transferring the present model to general studies on protein dynamics

The new analysis methods developed here broaden the applications of FRAP. Specifically, our modelling overcomes low signal-to-noise ratios and accounts for diffusion during intentional

bleaching and unintentional bleaching during imaging. Bleaching due to imaging is here used to analyse binding dynamics. Indeed, at the high laser powers needed in vivo a specific bleaching step may not be necessary, or even optimal. These advances may be generally valuable for studies of embryos and thick tissues or biomaterials by allowing higher laser powers. Also, our results show that pooling results from different cells in a complex in vivo environment may lead to error and mask individual cell-to-cell variability. Finally, our model for one-dimensional diffusion may be especially useful in elongated cells such as muscle cells or neurons. The model was implemented in open-source software that fits diffusion coefficients to data without programming ('Materials and methods'). Our methodology may have general application for analysis of protein dynamics in vivo.

In conclusion, the present study reveals for the first time the complex dynamics of Dystrophin in maturing muscle cells within the intact animal. It reveals important cell-to-cell variations that most likely reflect fibre or tip maturation but could have another origin. Both developmental state and genetic background are thus expected to influence the stability of Dystrophin, which could prove important in the clinic. Unstable binding or overall shortage of a specific pool may affect Dystrophin turnover and muscle performance. In future, the methodology developed here can be used to test for the comparative performance of short Dystrophin forms in use in gene therapy trials, with the aim of focusing on stable versions that may favour a more successful clinical outcome.

## Materials and methods

### Expression plasmids

Full-length 427-kd human Dystrophin (huDys) was produced by generating a human Dystrophin cDNA using long-range PCR (primers F1: SpeI_GACTAGTGTGTTCTTCATATGTATATCCTTCC; R1: MluI_CGA CGCGTCATTGTGTCCTCTCTCATTG), digested with SpeI and MluI and cloned into pCI plasmid (Promega, Madison, WI, United States) downstream of a CMV promoter at the NheI and MluI restriction sites. Insertion of GFP tag: (1) primers F2 (TCACCTCGAGAAAGTCAAGGCACTTCGAGGAGAAATTG, matching the 3′ huDystrophin cDNA plus a 5′ XhoI site) and R2 (CCTCGCCCTTGCTCACCATGGTTGTGGCCATTGTG TCCTCTCTCATTGGCTTTCCAGGGGTATTTCTTC, designed to remove the Dystrophin stop codon and harbouring first 30 nucleotides of eGFP cDNA) were used on huDys; (2) eGFP cDNA was amplified with F3 (GAAGAAATACCCCTGGAAAGCCAATGAGAGAGGACACAATGGCCACAACCATGGTGAGCAAGGGC GAGG, containing a 5′ free tail encoding the Dystrophin cDNA end) and R3 (GGTACCACGCGTTTACT TGTACAGCTCGTCCATGCC, plus a MluI site); (3) finally, the two products were mixed, amplified with F2 and R3, digested with XhoI and MluI and inserted into pre-digested huDys to generate huDysGFP. GFP was expressed from pCMV-GFP (Addgene 11153). Full-length zebrafish Dystrophin (*Lai et al., 2012*) GFP tagged was synthesized by GenBrick and subcloned into pCI-Neo at the MluI-SalI site (GenScript USA Inc., Piscataway Township, NJ, United States). All constructs were fully sequenced.

### Animals, injections, heat shock and embedding

Fish used were King's wild-type *Danio rerio*, dmd$^{ta222a/+}$, *Tg(actc1b:mCherry)$^{pc4}$* (*Cole et al., 2011*), and *Gt (dmd-Citrine)$^{ct90a}$* (*Trinh et al., 2011*; *Ruf-Zamojski et al., 2015*) and were staged and reared as described (*Westerfield, 1995*). Plasmids were injected into 1-cell stage embryos at 20–40 pg/embryo. Phenolthiourea (0.003%) was added to inhibit pigmentation. Heat shock was performed at 6 s and embryos analysed at 48 hpf (*Ruf-Zamojski et al., 2015*). To image, 48 hpf dechorionated embryos were anaesthetized with tricaine (0.2 mg/ml) and embedded in 1.5% low melting point agarose diluted in fish water.

### Immunohistochemistry and in situ hybridization

Standard protocols were used. Embryos were fixed in cold methanol for Dystrophin staining, or otherwise in paraformaldehyde 4%. Antibodies were mouse anti-Dystrophin MANDRA1 (1:100; Novocastra, Roche, Basel, Switzerland), mouse anti-human Dystrophin Dy8 (1:100; Novocastra, Roche), rabbit anti-GFP (1: 500; Roche), goat anti-mouse Alexa-543, and goat anti-rabbit Alexa-488. NMJ were detected with conjugated bungarotoxin-594 (1:1000, Invitrogen, ThermoFisher Scientific, Waltham, MA, United States). In situ hybridization was performed as previously described (*Hinits and Hughes, 2007*), with a specific probe against human Dystrophin spectrin repeats 20–22.

## Microscopy and analysis software

An upright Zeiss Exciter laser scanning microscope (LSM) with a 40×/1.1 W Corr LD C-Apochromat objective, and an inverted Zeiss 710 LSM with a 20×/1.0 W Plan-Apochromat and a 40×/1.3 Plan-Apochromat objective were used for FRAP and Z-stacks. Acquisition and maximum intensity projections were made with ZEN 2009/2010 (Zeiss, Jena, Germany). Velocity version 6.0.1 (PerkinElmer, Waltham, MA, United States) was used for XYZ projection, to which a fine Gaussian filter was applied and brightness corrected for visualization purposes only. Images were uniformly contrasted with Adobe Photoshop CS4. Illustrations were made in Adobe Illustrator CS3. GraphPad Prism 6 was used for statistical analysis and graph plotting.

## Integrated density analysis

This parameter is a measure of the amount of fluorescence signal resulting from the expression of huDysGFP in a cell or defined sub-cellular region. Confocal Z-stacks of muscle fibres were acquired as 8-bit greyscale images with a voxel size of 0.147 μm × 0.147 μm × 1 μm (x,y,z). ImageJ v1.45a was used for the next steps. The fibre tips or cytoplasm regions were manually delimited on sum projections of the pixel intensities over z-stacks. The raw integrated density (sum of the values of the pixels) in the tips and cytoplasm areas was measured and corrected for the average background. An approximate best correction for the contribution of cytoplasm signal at the tip region, typically at up a 45° angle to the field of view, was made by subtracting the corresponding cytoplasmic signal to half the tip area.

## FRAP parameters

Bleaching was performed at 100% intensity of an argon laser at 488 nm for GFP and 514 nm for Citrine. The acquisition region was a 300 × 60 pixel rectangle (44 μm × 8.8 μm), the interval between scanning rounds 0.2 s at a pixel dwell of 1.6 μs. For cytoplasmic studies, an open pinhole was used to optimize capture of the dim Dystrophin cytoplasmic signal, whereas a 1 Airy pinhole was generally used for GFP (except cells 6 and 7, *Table 1*). For studying the tips, the pinhole was set at 1 Airy to improve imaging resolution of the tip region and bleaching was performed on a 60 × 20 pixels rectangle with a single scan at pixel dwell of 12.8 μs, thus minimizing bleaching time. Conditions for zfDysCitrine FRAP were adapted to avoid depleting too much of the very low cytoplasmic pool. Also see *Figures 4, 6A* and *Table 1*.

## Modelling cytoplasmic FRAP experiments

The model considers diffusion in one dimension, X, of a single species in a spatially uniform background and includes bleaching by each imaging scan. The differential equation for the concentration of fluorescent protein $C_f(X, t)$ is

$$\frac{\partial C_f(X, t)}{\partial t} = D \frac{\partial^2 C_f(X, t)}{\partial X^2} - \beta \Theta_{FV}(X) \left( \sum_i \delta(t_i) C_f(X, t) \right),$$

where $D$ is the diffusion constant. The first two terms are the normal diffusion partial differential equation. The third term accounts for bleaching at each image acquisition. For each acquired image a fraction $\beta$ of the fluorescent protein in the imaged area is assumed bleached. The sum is over the image acquisition times $t_i$. The indicator function $\Theta_{FV}(X)$ is one within the scanned field of view and zero outside of it. The delta function at time $t_i$ of the acquisition of the $i$'th image ($\delta(t_i)$), assumes that bleaching occurs in the entire imaged region instantaneously. The fitting procedure is a least-squares two-parameter fit, $D$ and β, to the FRAP curve. The boundary conditions for this equation are that the first image after bleaching is given by a Gaussian fit to the profile in this first image, and zero-flux boundary conditions at the two tips of the model cell. The equation is solved within a box approximately as long as the cell and much larger than the imaged region. The entire cell is modelled as the protein diffuses in and out of the imaged region. The effect of varying cell length and position of the bleached region in the model was quantified and deemed minimal (*Table 1*).

To compare the experimental data with the model results, background subtracted pixel values are averaged along the $Y_T$-axis to obtain profiles of intensity as a function of $X_T$. Background is defined as the average intensity signal from a rectangular region outside the cell over pre-bleach images 4–20. A normalized profile for the first post-bleach point is constructed by dividing the first post-bleach profile

by the average of the profiles in pre-bleach time points 4–20. A Gaussian $A_0 - C_0 exp[-(X - X_0)^2/(2\sigma^2)]$ is fit to this normalized post-bleach profile, providing an initial profile for the FRAP-curve simulation. The parameters of the profile at the end of bleaching, $A_0$, $C_0$, $X_0$, and $\sigma$ are, respectively, the normalized intensity far from the region bleached, the maximum bleaching depth, the centre of the bleached region (along the $X$-axis), and the bleaching width. $A_0$ is set to $A_0 = 1$, the remaining three parameters are fit.

The normalized FRAP curve experimental points are compared with the average value in the computed profile in the bleached region. Fitting is done by minimizing the sum-of-the-squares of the difference between the two, to obtain the best-fit values of $D$ and $\beta$. For fits to the initial recovery only, bleaching is small and so a one-parameter fit to $D$ is done in this case.

## Analysis of fibre tip FRAP data

Fibre tips are divided into two $60 \times 20$ pixels areas, each covering approximately half of a typical tip, together with some cytoplasm and some pixels outside the cell. (*Figure 6A*). One box is intentionally bleached (bleached tip), while the other is imaged in an identical way but not intentionally bleached (unbleached tip). A region of similar size in the cytoplasm at a distance from the bright tip region is also analysed. For all regions, the background is subtracted and time points 4 to 20 are used to generate an initial pre-bleach average image for normalization, as for the cytoplasmic studies described above. The background-subtracted intensity is assumed to be proportional to Dystrophin concentration. To understand Dystrophin dynamics at the tip, and to distinguish between different bound populations, direct semi-quantitative analysis of bleaching and recovery is used. FRAP data are analysed in three ways:

1. Direct analysis of the FRAP curve for the bleached half of the tip. Lack of recovery is strong evidence of immobility on the timescale of the experiment. Rapid, but partial, recovery, are indicative signatures of a mobile pool and an immobile pool.
2. Comparative analysis of unnormalized intensities in the unbleached tip region and cytoplasm. Photobleaching due to imaging lowers the final intensity; this effect and a small immobile pool may be indistinguishable. To counter this problem, the identical photobleaching-due-to-imaging received by the unbleached tip region and cytoplasm is used to probe the dynamics. Unnormalized intensity plots (background-subtracted) allow direct comparison of the amount that is un-intentionally bleached and then recovers in the tip and the cytoplasm. Presumably, the size of the cytoplasmic component in a tip pixel is at most equal to that in a cytoplasmic pixel, and its FRAP dynamics are similar. Therefore, a permanent drop in tip intensity that is much larger than the drop in the cytoplasmic intensity indicates a large immobile pool. In addition, a dip for the tip signal that is much larger than that in the cytoplasm, and that rapidly recovers, indicates that there is a dynamic bound pool at that tip. Analysis of unnormalized minus cytoplasm intensity curves evaluates if at least 50% of the final recovery occurred at the first or second time points after switching from fast to slow (every 10 s) acquisition rates.
3. Analysis of the difference between bleached and unbleached tip regions. Both regions of the tip receive the same bleaching-due-to-imaging for ~250 s. If the final difference is large, presumably there is a bound population with a bound lifetime of at least hundreds of seconds. However, if the difference is close to zero, then any bound species is dynamic on this timescale.

## Whole cell bleaching and evaluation of recovery fraction due to dark state

Tests were made to evaluate whether bleaching may cause a significant portion of huDysGFP to enter a transient dark state, which then contributes to the fractional recovery after photobleaching. Typically, huDysGFP was bleached in an entire muscle fibre in vivo. Cells expressing high levels (including in the cytoplasm) were chosen to allow better detection of potential low levels of a dark-state pool. Bleaching was performed using the Argon laser at 100%, at which intensity we measured a direct laser power of 0.39–0.5 mW. To image the whole cell, a 20×/1.0 W Plan-Apochromat (Zeiss) objective was used. Due to the size of a muscle cell, similar tests bleaching the entire cell are not possible to perform using exactly the same conditions as in our FRAP experiments, where a 40×/0.8 Achroplan (Zeiss) objective was used. However, the laser power per area is higher in the conditions we use both in cytoplasmic and tips FRAP experiments, which is demonstrated by *Mueller et al. (2012)* to further decrease the effect of dark state. After whole-cell bleaching very low recovery after photobleaching is detected (<1%), presumably due to a shift from dark state to excitable huDysGFP.

## Software

A user-friendly application to analyse cytoplasmic diffusion along the long axis of a cell is freely available (see instructions in *Bajanca et al., 2015*).

## Datasets

The original datasets and main individual FRAP analysis files are deposited in the Dryad Digital Repository (*Bajanca et al., 2015*).

## Acknowledgements

We thank Glen Morris for anti-Dystrophin antibodies, Roland Roberts for cDNA probes, Tapan Pipalia for counting cells, Rainer Heintzmann for extensive and generous help on imaging, Jeff Chamberlain for sharing constructs and advice, Le Trinh for sharing *Gt(dmd-Citrine)*[ct90a] line.

## Additional information

### Funding

| Funder | Grant reference | Author |
|---|---|---|
| Association Monégasque Contre les Myopathies | ICE consortium | Luis Garcia, Simon M Hughes |
| Région Midi-Pyrénées | 13053025 | Fernanda Bajanca, Eric Theveneau |
| Fondation pour la Recherche Médicale (FRM) | AJE201224 | Eric Theveneau |
| European Commission (EC) | Marie Curie MC253305 | Fernanda Bajanca |
| Kwan Trust | | Vinicio Gonzalez-Perez |
| University of Surrey | Overseas Research Scholarship | Vinicio Gonzalez-Perez |
| Engineering and Physical Sciences Research Council (EPSRC) | Vacation Bursary | Sean J Gillespie |
| Medical Research Council (MRC) | Programme grant G1001029 | Simon M Hughes |

The funders had no role in study design, data collection and interpretation, or the decision to submit the work for publication.

### Author contributions

FB, Conception and design, Acquisition of data, Analysis and interpretation of data, Drafting or revising the article; VG-P, Developed the model; SJG, Wrote the software; CB, Made the human Dystrophin constructs; LG, Conception and design, Contributed unpublished essential data or reagents; ET, Analysis and interpretation of data, Drafting or revising the article, Contributed unpublished essential data or reagents; RPS, SMH, Co-last author, Conception and design, Analysis and interpretation of data, Drafting or revising the article

## Additional files

### Major dataset

The following dataset was generated:

| Author(s) | Year | Dataset title | Dataset ID and/or URL | Database, license, and accessibility information |
|---|---|---|---|---|
| Bajanca F, Gonzalez-Perez V, Gillespie SJ, Beley C, Garcia LM, Theveneau E, Sear RP, Hughes SM | 2015 | Data from: In vivo dynamics of skeletal muscle Dystrophin in zebrafish embryos revealed by improved FRAP analysis | https://dx.doi.org/10.5061/dryad.qg8dt | Available at Dryad Digital Repository under a CC0 Public Domain Dedication. |

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
