## [Decision Letter]

[Editors’ note: this article was originally rejected after discussions between the reviewers, but the authors were invited to resubmit after an appeal against the decision.]

Thank you for choosing to send your work entitled “In vivo dynamics of muscle Dystrophin revealed by improved FRAP analysis” for consideration at *eLife*. Your full submission has been evaluated by Sean Morrison (Senior Editor) and three peer reviewers, one of whom is a member of our Board of Reviewing Editors, and the decision was reached after discussions between the reviewers. Based on our discussions and the individual reviews below, we regret to inform you that your work will not be considered further for publication in *eLife*.

Although all reviewers concur on the novelty and potential relevance of your work, major concerns were raised on the experiments performed to support your conclusions. In particular the over-expression of human dystrophin in a wt zebrafish, which already produces its own dystrophin, was considered as a potential cause for the cytoplasmic pool. Moreover, the differences between the human and the fish protein may be at the basis of the different membrane bound pools.

Reviewer #1:

This is a well written and executed work that proposes a novel, exciting and potentially very important new mechanism of dystrophin localisation and dynamics inside the muscle fibre. Given the relevance of the conclusions deriving from this work, it is essential that possible biological artefacts be ruled out. Unfortunately they are not.

The major concern is the fact that a human protein is over-expressed in a remotely distant species. Therefore to validate the conclusions, it would be important to know:

1) What are the level of expression of total dystrophin (zebrafish + human) in the fish and how many times it exceeds the wt level? A WB analysis of total dystrophin would have clarified the level of overexpression. The cytoplasmic pool may be simply a consequence of the over expression and may not exist in a wt animal. Moreover, in order to conclusively demonstrate that human dystrophin functionally replaces the fish counterparts, experiments should have been performed in dystrophic zebrafish embryos, with the double advantage to be able to modulate the level of expression to wt levels and, equally important, to demonstrate that human dystrophin rescues muscular dystrophy in the fish. The authors mention that human dystrophin rescues muscular dystrophy in the fish only as “unpublished” in the Discussion. While it is understandable that they may want to publish these data separately, the problem of level of expression remains.

2) Even though human dystrophin localises like its zebrafish counterpart, the spatial organization is significantly different in the embryos of two species as is the overall muscle structure. Moreover, accumulation at the muscle fibre tip is not observed in human muscle and this is a matter of concern to extrapolate that what observed in the zebrafish also occurs in human muscle. To conclusively demonstrate this, the experiments should be repeated in a well-differentiated DMD muscle fibre in vitro, as it is now possible to obtain with biomaterials and stretching. While this would be a completely new work and it would be unfair to ask to include all this work in this paper, the concern on extrapolating these data to human muscle remains and bears consequences on the design of dystrophin replacement strategies in patients.

3) Although all the proteins interacting with dystrophin are well conserved during vertebrate evolution, there is no evidence in the paper that rules out that the bound mobile dystrophin pool may be the consequence of a weaker interaction of human dystrophin with fish complex proteins and may not exist in human cells, or even in the zebrafish embryo expressing a wt level of the fish protein.

Because of these major biological concerns, publication of the article in its present form is not recommended, since the intriguing and important conclusions of this work are not validated by the data presented.

Reviewer #2:

The authors propose a novel hypothesis according to which a large fraction of Dystrophin is present in a soluble cytoplasmic form. However, the whole study is based on unsupported and possibly erroneous assumptions that human Dystrophin, fused to GFP tag, and over-expressed in zebrafish myofibres, is functionally and kinetically equivalent to the native protein.

First, it is important to consider that zebrafish and human Dystrophin differ at amino acid level and hence the affinity of the human protein might be lower to its binding partners at the zebrafish sarcolemma. This can lead to the accumulation of the human protein in the zebrafish myofibre cytoplasm. Indeed, the data presented in the manuscript tends to support this, as the authors did not detect any endogenous zebrafish Dystrophin in the cytoplasm. It might be due to undetectable level in the cytoplasm, as the authors hypothesize. However, if this is the case, then it should be proven by cell fractionating and biochemical methods.

Second, all the conclusions rely on using an overexpression plasmid that is controlled by viral promoter. It is probable that there are limited binding sites for Dystrophin at the cell membrane and such overexpression system would lead to the accumulation of protein in the cytosol. In fact, the authors show that at low expression level they could not find Dystrophin-GFP in the cytoplasm. Expression driven by endogenous promoter would have eliminated all these concerns (i.e. BAC transgenic fish).

Third, the authors fail to show whether the C-terminal GFP tag alters Dystrophin kinetics. Although GFP size is small relative to Dystrophin, it is large enough to alter binding affinity to other proteins. While GFP-tagged human Dystrophin can rescue dystrophic zebrafish in short-term experiments, then this do not require native protein kinetics. Control experiments with N-terminal tag would have been essential. Especially considering that the C-terminus of Dystrophin is located towards the membrane bound dystroglycan complex and the addition of GFP could have weakened the interaction of the fusion protein with it.

Fourth, as it appears from the Materials and methods section, the area that was bleached in FRAP was significantly lager in the cytoplasm than at the membrane. Yet, one cannot compare two regions with different size as diffusion occurs faster in a smaller area. By using a small bleached area at the membrane one would overestimate protein diffusion there in comparison to much larger bleached area in the cytoplasm.

In sum, it is possible that Dystrophin can be present in the cytoplasm, but the manuscript fails to provide convincing experimental support for this. Correct analysis of Dystrophin kinetics in zebrafish would require complete restructuring of the work, focusing on the kinetics of the native protein, preferably in its original genomic location. Biochemical and cell biology experiments can be implemented to study Dystrophin in human muscle fibres. In either case this would involve a significant amount of work and this reviewer cannot see a possibility to recommend the current study for publication.

Reviewer #3:

This manuscript describes a very interesting investigation of the mobility of dystrophin within developing muscles of the zebra fish embryo. Although the functional equivalencies of this protein in the two species are not precisely matched, the visual accessibility provided by the zebra fish system provide a unique opportunity to examine the behaviour of this protein at the molecular level. The general validity of the approach is justified by a comparison of the distribution of the labelled human dystrophin with that of the endogenous zebra fish dystrophin. The investigation of the mobility of labelled dystrophin protein is a heavily modified version of the standard recovery from bleaching technique for measuring protein mobility.

The outcome is the conclusion that there is little lateral diffusion or other movement within the membrane-bound dystrophin compartment, most restoration of dystrophin to sites of bleached membrane occurring by interchange with the much larger cytoplasmic compartment. The other main conclusion is that there are two bound states of dystrophin, a strong slowly exchanging state, presumed to be in strong association with other proteins of the membrane complex and a more rapidly exchanging state the nature of whose binding is uncertain.

Overall, the investigation is interesting, carefully performed, controlled and interpreted. It provides new information that, with due allowance for the differences between the model of developing zebra fish muscle and postnatal human muscle in which it the object of medical interest, usefully enhances our understanding of its functional properties.

[Editors’ note: what now follows is the decision letter after the authors submitted for further consideration.]

Thank you for your letter appealing our initial decision. We have consulted among the editors and would be willing to encourage the submission of a revised manuscript, though we cannot guarantee the final outcome.

In your letter you stated two main points that addressed two of three major concerns. However we believe that you may also have or can easily produce data to address the third and final important concern. Specifically:

1) You stated clearly (but this was not in the first version) that the cytoplasmic pool might well be an artifact due to over expression. Once this is made clear there is no longer the risk to mislead the reader about a physiologically significant pool of soluble dystrophin.

2) You proposed to insert the rescue of the dystrophic fish by human dystrophin. A careful analysis of the resulting phenotype would strengthen the work considerably.

3) The third concern that you do not address in your letter relates to possible different binding kinetics between fish and human proteins. This is important because the labile membrane bound pool may be a consequence of this difference. If you have performed, as control, rescue of dystrophic fish by fish dystrophin and you also see the two pools, that would be sufficient to rule out a species difference. Ideally one would like to see the same also in human cells but we understand that this would be a whole new set of experiments that you may not be in a position to do. If the fish data can be provided, then just a word of caution in the discussion, as to the fact the final confirmation of the two pools in human muscle awaits experiments in human cells should acknowledge the unlikely possibility that things may be different in human.

Therefore, if you think you can address the three major points raised we would be happy to reconsider the manuscript.

---

## [Author Response]

*[Editors’ note: this article was originally rejected after discussions between the reviewers, but the authors were invited to resubmit after an appeal against the decision*.*]*

We are pleased to read that the reviewers agree on the novelty and potential relevance of our work, and understand the major concerns that lead to the rejection decision. However, each of the major concerns is based on a wrong assumption that we would like to challenge.

Firstly, it is not true to say that zebrafish Dystrophin is located differently from human Dystrophin and that this gives grounds to worry about the relevance of our work to humans. We present below extensive bibliographic evidence of the similar distribution of human and zebrafish Dystrophins. While, of course, work in a model organism cannot be thoughtlessly applied to humans, it has nevertheless proved highly relevant in many areas of biomedicine. Secondly, it is incorrect to suppose, as Reviewer 2 seems to do, that we regard the observed cytoplasmic pool of Dystrophin as a key feature of our findings. We obviously did not explain clearly enough, for which we apologize. Like Reviewer 2, we think the large cytoplasmic pool present in some fibres is simply an artefact of over-expression. Nevertheless, we showed that the size of the cytoplasmic pool does not determine Dystrophin binding dynamics, and specifically, does not determine our new weakly membrane-bound Dystrophin pool. Thirdly, Reviewers 1 and 2 both thought we needed to demonstrate functional relevance. We now have data showing that human Dystrophin-GFP rescues the fish *dmd* mutant phenotype cell autonomously at a single fibre level. While these new extended functional data were previously intended to form a second manuscript, we would be willing to add them to the present manuscript if a resubmission were allowed.

Please find below a more detailed refutation of the major criticisms by reviewers 1 and 2 that includes new data to support our claims. Reviewer 3 is very positive and does not present any major criticism.

Reviewer #1:

*1) What are the level of expression of total dystrophin (zebrafish + human) in the fish and how many times it exceeds the wt level? A WB analysis of total dystrophin would have clarified the level of overexpression*.

Thank you for this comment. We have addressed this point. Unfortunately, western blots are not relevant as our expression is intentionally mosaic in single isolated cells to permit FRAP. In addition, note that immunostaining using MANDRA1 antibody, which recognizes a conserved epitope on both human and zebrafish Dystrophins, does not show increased accumulation at the ends of fibres containing huDysGFP compared with adjacent fibres lacking huDysGFP (Figure 1). Thus, the presence of excess cytoplasmic huDysGFP likely displaces some of the endogenous Dystrophin, rather than forcing extra binding sites to form. Further, the huDysGFP intensity in the whole fibre and locally at the fibre tips was measured in wild-type embryos (where it coexists with the endogenous Dystrophin) and in *dmd*^*ta222a*^ dystrophic embryos (Figure 8). In the absence of competition with the endogenous protein, huDysGFP can occupy all the available binding sites, showing a 2.5 fold increase in the intensity ratio tips:cytoplasm comparing with that found in wild-type background. Or, in different words, fibres expressing the same intensity in the cytoplasm can accumulate more at the fibre ends in the mutant background. This nicely shows that our experiments in the wild type background were made in non-saturating conditions. Moreover, we have repeated the key FRAP experiments in the dystrophic background and our analysis found no differences regarding the cytoplasmic pool or the mobile and immobile bound fractions (Figure 8; Results section “Human Dystrophin efficiently rescues zebrafish dystrophic embryos and two bound pools are still found in the absence of competition with endogenous Dystrophin”).

*The cytoplasmic pool may be simply a consequence of the over expression and may not exist in a wt animal*.

We do agree that most of the cytoplasmic Dystrophin that accumulates in highly expressing fibres is a consequence of the over expression. We now make this point clear both in the Results and Discussion sections. However, the existence of a cytoplasmic pool does not invalidate our results. Firstly, our data rules out that the size of the cytoplasmic pool affects the overall binding (see Results section “Increase of cytoplasmic Dystrophin does not affect accumulation at the fibre tips”). Secondly, the weakly-bound Dystrophin pool can be found in fibres with no detectable or very low levels of cytoplasmic unbound huDysGFP (see Figure 6; Table 2), and cells with high cytoplasmic huDysGFP intensity do not necessarily recover more (compare Figure 6; Table 2). This implicates that the size of the cytoplasmic pool does not determine the weakly-bound Dystrophin pool.

Furthermore, we have now added the FRAP analysis of endogenously-driven zebrafish Dystrophin (Figure 10; Results section “Endogenous zebrafish Dystrophin diffusion and binding dynamics”). Our results show clearly that there are low levels of endogenous cytoplasmic Dystrophin in muscle fibres of *Gt(dmd-citrine)*^*ct90a*^ zebrafish embryos, in which the endogenous gene was tagged by a gene trap. We demonstrate that endogenous zebrafish Dystrophin also shows the characteristic two bound pools at the membrane.

Importantly, note that the observed accumulation in the cytoplasm mimics the expression pattern observed in embryonic/foetal human muscle fibres. Wessels and colleagues showed that in human embryos Dystrophin first appears in the *cytoplasm*, at the *ends of myotubes* (see Figure 3 in [65], ). It then becomes widespread throughout the myofibres in foetal stages (see Figure 4 in [65]), becoming restricted to the sarcolemma only in maturing and adult muscles. Several other studies from different labs and using a wide range of different antibodies (including some of the most extensively used nowadays like DYS-1, DYS-2, DYS-3) confirm the initial cytoplasmic accumulation in human embryos and that the level of internal staining declines with age and is considerably lower in adult muscle ([18]; see Figure 3 in [16]; [47]; [59]). Interestingly, a cytoplasmic Dystrophin pool was also found in regenerating muscle fibres (e.g., Kääriäinen et al., Neuromuscul Disord. 2000), in differencing human primary muscle cultures (e.g., Miranda et al., Am J Pathol. 1988) and in the adult heart (e.g, Peri et al., Mol Cell Biochem. 1994). Therefore, even though there are cases in our study where large amounts of Dystrophin accumulate in the cytoplasm due to overexpression, it is very likely that baseline biological levels are present in most cells, at least in the embryonic context that we study here.

*Moreover, in order to conclusively demonstrate that human dystrophin functionally replaces the fish counterparts, experiments should have been performed in dystrophic zebrafish embryos, with the double advantage to be able to modulate the level of expression to wt levels and, equally important, to demonstrate that human dystrophin rescues muscular dystrophy in the fish. The authors mention that human dystrophin rescues muscular dystrophy in the fish only as “unpublished” in the Discussion. While it is understandable that they may want to publish these data separately, the problem of level of expression remains*.

Thank you for pointing this out. We are happy to include the analysis of human Dystrophin expression in dystrophic fish (*dmd*^*ta222a/ta222a*^). We show that exogenous human Dystrophin efficiently rescues the dystrophic phenotype indicating that human Dystrophin is fully functional within fish muscle cells and analyse Dystrophin binding dynamics in rescued DMD fibres by FRAP (Figure 8; Results section “Human Dystrophin efficiently rescues zebrafish dystrophic embryos and two bound pools are still found in the absence of competition with endogenous Dystrophin”). We find no differences between the dynamics of huDysGFP in wild type embryos and in DMD embryos. This result indicates that the competition with endogenous zebrafish Dystrophin in the original wild-type host embryos was not the cause of the mobile-bound Dystrophin pool we observed.

*2) Even though human dystrophin localises like its zebrafish counterpart, the spatial organization is significantly different in the embryos of two species as is the overall muscle structure. Moreover, accumulation at the muscle fibre tip is not observed in human muscle and this is a matter of concern to extrapolate that what observed in the zebrafish also occurs in human muscle. To conclusively demonstrate this, the experiments should be repeated in a well-differentiated DMD muscle fibre in vitro, as it is now possible to obtain with biomaterials and stretching. While this would be a completely new work and it would be unfair to ask to include all this work in this paper, the concern on extrapolating these data to human muscle remains and bears consequences on the design of dystrophin replacement strategies in patients*.

This is indeed a very interesting point. However, we respectfully disagree with the reviewer argument that there are key differences between our observations and the reality of Dystrophin localisation in human muscles that render our data on human Dystrophin in fish cells irrelevant. In the original submission, it was our mistake not to properly highlight the similarity between human and zebrafish Dystrophin localisations, when both cell location and developmental stage are taken into account. Human Dystrophin localises in zebrafish embryonic muscles just as it localises in human embryonic muscles. As already mentioned above, several studies showed that in human embryos and foetuses Dystrophin first appears in the *cytoplasm*, at the ends of myotubes and then widespread, before becoming restricted to the sarcolemma (see Figure 3 in [65]; [18]; see Figure 3 in [16]; [47]; [59]). In the original study, [65] said that “The sarcoplasmic localization in embryonic and fetal tissue and the sarcolemmal localization of dystrophin in mature muscle *suggests the accumulation of dystrophin in the cytoplasm prior to its integration into the membrane*.” The same group also detected cytoplasmic accumulation in an aborted Duchenne foetus, with antibodies against N-terminal Dystrophin (Ginjaar et al., 1989). While at the time they did not have the tools available to test their hypothesis, our results confirm the existence of a cytoplasmic pool of dystrophin in embryonic muscle cells. It would be nice to be able to analyse this cytoplasmic pool directly in human muscle cells. We now mention this point in the Discussion. We believe it is advantageous to analyse Dystrophin dynamics in developing muscle fibres in the embryo, where binding complexes are actively forming. In the adult, there are presumably more stable complexes with a slower turnover and the discrete details of the assembly mechanism may become harder to spot. However, in human DMD therapy situations the complexes will be forming de novo in existing and nascent regenerating fibres.

Regarding regenerating fibres, relevant in the DMD context, it was shown that the localisation of Dystrophin *recapitulates that found during embryonic muscle development* during regeneration of rat muscles in vivo (32) and again in differentiation of human primary muscle cultures (45; 59). In both cases Dystrophin starts by localising in the cytoplasm before incorporating the membrane complexes. Moreover, Dystrophin *was found enriched at the at the myotendinous junction (MTJ) localised at the fibre tip region in human muscle biopsies* (see Figure 6 in [69]; Note the antibodies used are still among the most widely used to these days) and in *several other mammals*, namely mouse (see Figure 5 in [56]), rat (32), and guinea pig (see Figure 7 in [44] ). The fact that most studies focus on the costameres expression (which is commonly all that can be observed in a biopsy as surgeons do not perform biopsies near tendons) may in fact contribute to an incomplete view of the biology of Dystrophin in human cells. However, we would like to stress that as our data stands, we do not make direct extrapolations to the human muscle or DMD patients, but the working model arising from our data raises new questions and opens a new avenue for more applied research to follow. We added a statement to the Discussion acknowledging the importance of confirming the dynamics of human Dystrophin in human cells in future studies. We also integrate better the literature on dystrophin in human cells to make it clearer that what we observe in our experimental system fits with previous studies in embryonic and regenerating human muscles (Introduction and Discussion sections).

*3) Although all the proteins interacting with dystrophin are well conserved during vertebrate evolution, there is no evidence in the paper that rules out that the bound mobile dystrophin pool may be the consequence of a weaker interaction of human dystrophin with fish complex proteins and may not exist in human cells, or even in the zebrafish embryo expressing a wt level of the fish protein*.

We completely agree with this comment and have addressed it in two ways. First, we made a new expression construct to exogenously express GFP-tagged zebrafish Dystrophin. This construct was expressed in both wild-type and dystrophic embryos and the binding dynamics were analysed (Figure 9; Results section “Two bound populations also characterize the dynamics of zebrafish Dystrophin-GFP”). The two bound pools are present in both genetic backgrounds, confirming that the mobile-bound pool is not a consequence of a different binding kinetics between human and zebrafish proteins. However, there was still the possibility that the mobile-bound pool was due to the over-expression strategy. Therefore, our second approach was to perform FRAP analysis of endogenously-driven zebrafish Dystrophin As mentioned in the reply to point 1, this mobile-bound form is also found in muscle fibres of *Gt(dmd-citrine)*^*ct90a*^ zebrafish embryos (Figure 10; Results section “Endogenous zebrafish Dystrophin diffusion and binding dynamics”). We believe that these new experiments, together with the clear acknowledgment of the importance of confirming the dynamics of human Dystrophin in human cells in future studies that we now add to the Discussion bring our manuscript closer to the expectations of the reviewer.

Considering this is the first study of Dystrophin dynamics in vivo we cannot know if Dystrophin behaves similarly in all species and all developmental stages, but the question is raised. The FRAP analysis developed for this work can now be used to analyse human Dystrophin dynamics in primary muscle cell cultures, or even pluripotent human stem cells differentiated into muscle fibres. However, to perform these experiments in human cells is no trivial task. It will be necessary to transfect the full-length human Dystrophin, or to modify the endogenous *dmd* locus to insert a fluorescent tag, and then to study well-differentiated muscle fibres in vitro, for example by using biomaterials and stretching techniques as suggested by this reviewer. Ideally, 3D cultures with muscle-tendon attachments and innervations would guarantee more reliable results. We believe this first study of Dystrophin dynamics on the zebrafish embryo shows thought-provoking results and sets up a methodology that may now be used to explore further the implications of the proposed model.

Taking into consideration this evidence, we believe that we now provide the main technical controls asked by the reviewer. Importantly, we also better integrate the literature on dystrophin in human cells to make it clearer that what we observe in our experimental system fits with the proposed dynamics of dystrophin in embryonic and regenerating human muscles.

Reviewer #2:

*The authors propose a novel hypothesis according to which a large fraction of Dystrophin is present in a soluble cytoplasmic form*.

Thank you for this comment, which highlighted the fact that we had been unclear in our original submission. We do not regard the observed cytoplasmic pool of Dystrophin as a key feature of our findings. We think the large cytoplasmic pool present in some fibres is simply an artefact of over-expression. This is now made clear both in the Results and Discussion sections. Yet, since this large cytoplasmic pool is present we had to take it into account in the analysis. We show that this pool exist in a smaller scale in the endogenous situation (citrine fish, please see below) and that the two membrane form of dystrophin are still detectable regardless of the amount of cytoplasmic dystrophin present in the cells. We made several changes in the manuscript to make this point clear. We hope this reviewer will find it satisfactory.

*However, the whole study is based on unsupported and possibly erroneous assumptions that human Dystrophin, fused to GFP tag, and over-expressed in zebrafish myofibres, is functionally and kinetically equivalent to the native protein*.

These are fair concerns. We have now added several controls to better consolidate our results on the human Dystrophin. Importantly, we added data showing that human Dystrophin fully rescues the dystrophic phenotype (Figure 8; Results section “Human Dystrophin efficiently rescues zebrafish dystrophic embryos and two bound pools are still found in the absence of competition with endogenous Dystrophin”) indicating that huDys-GFP is functional in the fish background. In addition, we have analysed the dynamics of exogenous zebrafish Dystrophin with C-terminal GFP-tag, equivalent to the human Dystrophin-GFP, and compared both with endogenous Dystrophin of *Gt(dmd-citrine)*^*ct90a*^ zebrafish embryos, in which a Citrine-tag was inserted close to the actin binding site by a gene trap. We have found similar dynamics in all three conditions, despite the different tags and their different localisation within the protein. Further, our results show that exogenous zebrafish and human Dystrophin have diffusion and binding dynamics similar to those of endogenous zebrafish Dystrophin, in spite of the artificially raised cytoplasmic levels caused by over-expression. Therefore, neither the over-expression nor different binding kinetics between fish and human proteins affect Dystrophin dynamics in our system. In addition to the data added to the Results section (“Two bound populations also characterize the dynamics of zebrafish Dystrophin-GFP” and “Endogenous zebrafish Dystrophin diffusion and binding dynamics”; Figures 9 and 10), we have improved the Discussion to make clear that our system can be used to study human Dystrophin dynamics, with limitations (see “Validating Dystrophin over-expression in Zebrafish as a model system to study Dystrophin dynamics in vivo”). There, we stress the importance of confirming the dynamics of human Dystrophin in human cells in future studies.

Given that no similar analysis has ever been done on human protein in human (or any other species), our data does, at the very least, raise a valid issue.

*First, it is important to consider that zebrafish and human Dystrophin differ at amino acid level and hence the affinity of the human protein might be lower to its binding partners at the zebrafish sarcolemma. This can lead to the accumulation of the human protein in the zebrafish myofibre cytoplasm. Indeed, the data presented in the manuscript tends to support this, as the authors did not detect any endogenous zebrafish Dystrophin in the cytoplasm. It might be due to undetectable level in the cytoplasm, as the authors hypothesize. However, if this is the case, then it should be proven by cell fractionating and biochemical methods*.

The accumulation of Dystrophin in the cytoplasm does not invalidate our results, independently of any speculation about its driving source. Note that we thoroughly analysed the cytoplasmic pool to make sure its dynamics would be taken into account at all steps of the analysis. We have found that the overall binding of human Dystrophin at the membrane is not affected by the amount of Dystrophin in the cytoplasm (see Results section “Increase of cytoplasmic Dystrophin does not affect accumulation at the fibre tips”). In addition, the size of the cytoplasmic pool does not determine the presence of the mobile-bound Dystrophin pool. Indeed, our data show that the mobile-bound Dystrophin pool can be found in fibres with no detectable or very low levels of cytoplasmic unbound huDysGFP (Figure 6), and cells with high cytoplasmic huDysGFP intensity do not necessarily recover more (compare Figure 6).

Importantly, we have now added the FRAP analysis of endogenously-driven zebrafish Dystrophin (Figure 10; Results section “Endogenous zebrafish Dystrophin diffusion and binding dynamics”). Our results show that there are low levels of endogenous cytoplasmic Dystrophin in muscle fibres of *Gt(dmd-citrine)*^*ct90a*^ zebrafish embryos and we demonstrate that endogenous zebrafish Dystrophin also shows the characteristic two bound pools at the membrane. This observation is in line with previous observations. Cytoplasmic dystrophin was shown in developing human muscle fibres independently by different labs, using several different antibodies, many of which are still among the most widely used and more trusted ones (see Figures 3 and 4 in [65]
[18]; see Figure 3 in [16]; [47]; [59]). Cytoplasmic Dystrophin was also described in regenerating muscle fibres (32), in differencing human primary muscle cultures (45; 59) and in the adult heart (50). Considering that we express the human Dystrophin in a skeletal muscle embryonic environment it is not surprising, and it is even reassuring, that it would behave as in the human embryonic environment and be found in a cytoplasmic form. We apologize that the bibliographic evidence required to put many of our results in context was missing, which, together with the focus on comparing the exogenous human Dystrophin expression with that of the endogenous zebrafish Dystrophin may have misled the reviewer in many ways. We hope these points were now made clearer (Introduction and Discussion).

*Second, all the conclusions rely on using an overexpression plasmid that is controlled by viral promoter. It is probable that there are limited binding sites for Dystrophin at the cell membrane and such overexpression system would lead to the accumulation of protein in the cytosol. In fact, the authors show that at low expression level they could not find Dystrophin-GFP in the cytoplasm. Expression driven by endogenous promoter would have eliminated all these concerns (i.e. BAC transgenic fish)*.

We agree with the reviewer. The new analysis of endogenously-driven Dystrophine-Citrine eliminated these concerns.

*Third, the authors fail to show whether the C-terminal GFP tag alters Dystrophin kinetics. Although GFP size is small relative to Dystrophin, it is large enough to alter binding affinity to other proteins. While GFP-tagged human Dystrophin can rescue dystrophic zebrafish in short-term experiments, then this do not require native protein kinetics. Control experiments with N-terminal tag would have been essential. Especially considering that the C-terminus of Dystrophin is located towards the membrane bound dystroglycan complex and the addition of GFP could have weakened the interaction of the fusion protein with it*.

This is a fair point. We have compared the dynamics of zebrafish Dystrophin with a C-terminal GFP tag with that of zebrafish Dystrophin with Citrine tag localised close to the actin binding site (52). We found that Dystrophin dynamics is not affected by the position of the fluorescent suggesting that the tags are unlikely to significantly perturb the dynamics of Dystrophin.

In addition, note that we do show in the manuscript that human Dystrophin without the tag binds like huDysGFP (Figure 1). We also present new data to strengthen our technical controls on the functionality of the GFP-tagged human Dystrophin we use (Figure 8; Results section “Human Dystrophin efficiently rescues zebrafish dystrophic embryos and two bound pools are still found in the absence of competition with endogenous Dystrophin”). Hence, we believe that the risk of the GFP-terminal tag affecting the binding of Dystrophin is most likely negligible.

*Fourth, as it appears from the Materials and methods section, the area that was bleached in FRAP was significantly lager in the cytoplasm than at the membrane. Yet, one cannot compare two regions with different size as diffusion occurs faster in a smaller area. By using a small bleached area at the membrane one would overestimate protein diffusion there in comparison to much larger bleached area in the cytoplasm*.

We are puzzled by the reviewer’s claim that the diffusion rate would be faster in a smaller area. The diffusion rate is a measure of the distance travelled per unit of time (we measure it in micrometers travelled per second). By physical laws, the size of the observed region has no effect on the diffusion rate itself. Therefore, it is absolutely correct to compare different areas/volumes using diffusion equations.

Taking into consideration this evidence, we believe there is no reason to dismiss our results as pure artefacts. The new data presented provides further technical controls that we believe strengthen the manuscript. We acknowledge that a better integration of the literature on Dystrophin in human muscle cells was required to make it clearer that what we observe in our experimental system fits with the proposed dynamics of dystrophin in embryonic and regenerating human muscles.

*[Editors’ note: what now follows is the decision letter after the authors submitted for further consideration*.*]*

*1) You stated clearly (but this was not in the first version) that the cytoplasmic pool might well be an artifact due to over expression. Once this is made clear there is no longer the risk to mislead the reader about a physiologically significant pool of soluble dystrophin*.

We do agree that there is often an excess of cytoplasmic Dystrophin in the zebrafish host muscle fibres due to over expression. We now make this point clear both in the Results and Discussion sections. The question was raised by the referees with the concern that an excess of cytoplasmic Dystrophin could be the cause of the mobile-bound pool identified. To further support our original claims that the existence of extra cytoplasmic Dystrophin does not invalidate our results, we have now added the FRAP analysis of endogenously-driven zebrafish Dystrophin (Figure 10; Results section “Endogenous zebrafish Dystrophin diffusion and binding dynamics”). Although this experiment was not requested by the editors, we believe it adds important new data. Our results show clearly that there are low levels of endogenous cytoplasmic Dystrophin in muscle fibres of *Gt(dmd-citrine)*^*ct90a*^ zebrafish embryos, in which the endogenous gene was tagged by a gene trap. We demonstrate that endogenous zebrafish Dystrophin also shows the characteristic two bound pools at the membrane. With these new data in mind, the question is no longer whether there is a physiologically significant pool of soluble Dystrophin, but whether there is a similar low level of soluble Dystrophin in human cells. To put our data into context, we added citation of several independent studies suggesting that this may be the case, at least during development and regeneration. Whether the presence of this cytoplasmic pool is linked with the fact that Dystrophin complexes may be actively being formed in these situations remains an open and interesting question.

*2) You proposed to insert the rescue of the dystrophic fish by human dystrophin. A careful analysis of the resulting phenotype would strengthen the work considerably*.

We have now included the analysis of human Dystrophin expression in dystrophic fish (*dmd*^*ta222a/ta222a*^). We show that exogenous human Dystrophin efficiently rescues the dystrophic phenotype indicating that human Dystrophin is fully functional within fish muscle cells. In addition, we analysed Dystrophin binding dynamics in rescued DMD fibres by FRAP (Figure 8; Results section “Human Dystrophin efficiently rescues zebrafish dystrophic embryos and two bound pools are still found in the absence of competition with endogenous Dystrophin”). We find no differences whatsoever between the dynamics of huDysGFP in wild type embryos and in DMD embryos. This result indicates that the competition with endogenous zebrafish Dystrophin in the original wild-type host embryos was not the cause of the mobile-bound Dystrophin pool we observed.

*3) The third concern that you do not address in your letter relates to possible different binding kinetics between fish and human proteins. This is important because the labile membrane bound pool may be a consequence of this difference. If you have performed, as control, rescue of dystrophic fish by fish dystrophin and you also see the two pools, that would be sufficient to rule out a species difference. Ideally one would like to see the same also in human cells but we understand that this would be a whole new set of experiments that you may not be in a position to do. If the fish data can be provided, then just a word of caution in the discussion, as to the fact the final confirmation of the two pools in human muscle awaits experiments in human cells should acknowledge the unlikely possibility that things may be different in human*.

We made a new expression construct to exogenously express GFP-tagged zebrafish Dystrophin. This construct was expressed in both wild-type and dystrophic embryos and the binding dynamics were analysed (Figure 9; Results section “Two bound populations also characterize the dynamics of zebrafish Dystrophin-GFP”). The two bound pools are present in both genetic backgrounds, confirming that the labile-bound pool is not a consequence of a different binding kinetics between human and zebrafish proteins. However, there was still the possibility that the mobile-bound pool was due to the over-expression strategy. As mentioned in the reply to point 1, this mobile-bound form is also found in muscle fibres of *Gt(dmd-citrine)*^*ct90a*^ zebrafish embryos (Figure 10; Results section “Endogenous zebrafish Dystrophin diffusion and binding dynamics”).

We added a statement to the Discussion acknowledging the importance of confirming the dynamics of human Dystrophin in human cells in future studies . Considering this is the first study of Dystrophin dynamics in vivo we cannot know if Dystrophin behaves similarly in all species and all developmental stages, but the question is raised. The FRAP analysis developed for this work can now be used to analyse human Dystrophin dynamics in primary muscle cell cultures, or even pluripotent human stem cells differentiated into muscle fibres. However, to perform these experiments in human cells is no trivial task. It will be necessary to transfect the full-length human Dystrophin, or to modify the endogenous *dmd* locus to insert a fluorescent tag, and then to study well-differentiated muscle fibres in vitro, for example by using biomaterials and stretching techniques. Ideally, 3D cultures with muscle-tendon attachments and innervations would guarantee more reliable results. We believe this first study of Dystrophin dynamics on the zebrafish embryo shows thought-provoking results and sets up a methodology that may now be used to explore further the implications of the proposed model.